# Identifying latent state transitions in non-linear dynamical systems

**Çağlar Hızlı**
Aalto University
`caglar.hizli@aalto.fi`

**Çağatay Yıldız**
University of Tübingen
Tübingen AI Center

**Matthias Bethge**
University of Tübingen
Tübingen AI Center

**ST John**
Aalto University

**Pekka Marttinen**
Aalto University

## Abstract

This work aims to recover the underlying states and their time evolution in a latent dynamical system from high-dimensional sensory measurements. Previous works on identifiable representation learning in dynamical systems focused on identifying the latent states, often with linear transition approximations. As such, they cannot identify nonlinear transition dynamics, and hence fail to reliably predict complex future behavior. Inspired by the advances in nonlinear ICA, we propose a state-space modeling framework in which we can identify not just the latent states but also the unknown transition function that maps the past states to the present. Our identifiability theory relies on two key assumptions: (i) sufficient variability in the latent noise, and (ii) the bijectivity of the augmented transition function. Drawing from this theory, we introduce a practical algorithm based on variational auto-encoders. We empirically demonstrate that it improves generalization and interpretability of target dynamical systems by (i) recovering latent state dynamics with high accuracy, (ii) correspondingly achieving high future prediction accuracy, and (iii) adapting fast to new environments. Additionally, for complex real-world dynamics, (iv) it produces state-of-the-art future prediction results for long horizons, highlighting its usefulness for practical scenarios.

## 1 Introduction

We focus on the problem of understanding the underlying states of a dynamical system from its high-dimensional sensory measurements. This task is prevalent across various fields, including reinforcement learning (Hafner et al., 2019a) and robotics (Levine et al., 2016). For example, consider a cartpole, where instead of directly observing the underlying states (cart position and velocity, pole angle and angular velocity; Fig. 1 **(a)**), we observe a video stream of its behavior (Fig. 1 **(b)**). Our main objective is to learn latent representations and state transition functions from such high-dimensional sequences, which are useful for downstream tasks such as optimal control of the observed system.

Due to the partially observed nature of the problem, learning dynamics in the data space (e.g., pixel space) is not feasible, and previous works often focus on learning *latent dynamical systems* (Hafner et al., 2019b). However, such latent models commonly are not guaranteed to recover the true underlying states and transitions (*non-identifiability*), resulting in entangled representations, lack of generalization across new domains, and poor interpretability (Schmidhuber, 1992; Bengio et al., 2013).

**Identifiable representation learning** aims to address these challenges by learning the underlying factors of variation in the true generative model. To achieve this, our approach builds on the classical ideas of ICA, namely, the assumption of independent components. In the case of a linear emission function, this independence assumption has been used to identify states in dynamical systems (Ciaramella et al., 2006; Kerschen et al., 2007). For the nonlinear case, Hyvärinen & Morioka (2016; 2017) introduced an identifiability framework using two main assumptions: (i) latent states show non-stationarity or autocorrelation driven by an auxiliary variable (this is referred to as 'sufficient variability'), and (ii) given the auxiliary variable, they are conditionally independent. However, this framework assumes *mutually independent* latent states that do not affect each other (Hyvärinen & Morioka, 2016; 2017;

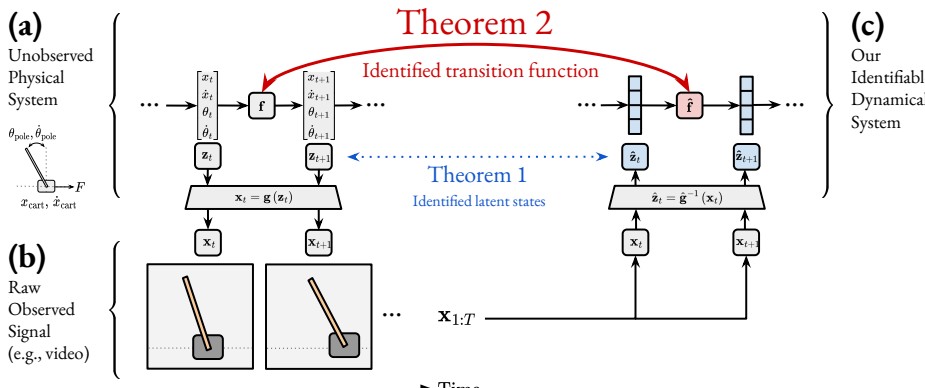

Figure 1: Sketch of our method and main theoretical contribution. **(a)** We assume an underlying *unobserved* dynamical system, e.g., a cartpole, where the full state $\mathbf{z}_t$ is composed of cart position and velocity and angle and angular velocity of the pole: $[x, \dot{x}, \theta, \dot{\theta}]$. **(b)** We partially observe the system as a sequence of video frames $\mathbf{x}_{1:T}$, which are used as input to our method. **(c)** We learn an inverse emission function $\hat{\mathbf{g}}^{-1} : \mathbf{x}_t \mapsto \hat{\mathbf{z}}_t$ that maps the raw observation signals to the estimated latent state variables, as well as a transition function $\hat{\mathbf{f}} : \hat{\mathbf{z}}_t \mapsto \hat{\mathbf{z}}_{t+1}$ that maps the past latent states to the present latent state. Identifiability of the latent states is ensured by Theorem 1. In addition to this, our main contribution is the identifiability of the transition function ensured by Theorem 2.

Hyvärinen et al., 2019; Khemakhem et al., 2020). This mutual independence assumption is unrealistic for dynamical systems, as the present state of the system depends on the past states, i.e., the transition function propagates the system state by nonlinearly mixing the past state components.

**Identifiability in dynamical systems.** Recently, Yao et al. (2021; 2022) extended identifiability theory to dynamical systems by replacing the assumption of *mutually-independent* states with *temporally-mixed* states, conforming to the same non-stationarity and autocorrelation conditions. Under the reformulated assumptions, they showed that it is possible to *identify* or recover the true latent states in a dynamical system (up to element-wise transformations). Using similar assumptions for an autoregressive process in the observation space, Morioka et al. (2021) identified the 'innovations' or process noise, which represents the stochastic impulses fed to the system. See Tab. 3 (App. A) for a comparison.

Although these approaches yield provably identifiable representations of the latent states, their theoretical frameworks fall short in identifying the underlying dynamics. To resolve this, inspired by nonlinear ICA, we introduce additional stochastic variables that are conditionally independent of each other given auxiliary variables. This conditional independence assumption implicitly encodes new, external information that is sufficiently independent of the observations (Hyvärinen et al., 2019, App. E). Interestingly, the idea of conditioning on auxiliary variables has been proposed in other contexts, e.g., Packard et al. (1980) showed that time-delayed auxiliary variables help resolve underlying data generating dynamics of chaotic systems from the residual independent noise.

**Our contributions.** We present the first framework that allows for the *identification of the unknown transition function* alongside latent states and the emission function[1] (see Fig. 1). Following previous works (Yao et al., 2021; 2022), we first establish the identifiability of the latent states (Fig. 1: Theorem 1). We show that identifying the transition function is ensured by further restricting the spaces of the process noise prior and the transition function (Fig. 1: Theorem 2). These restrictions form our main assumptions for identifiability: the process noise shows non-stationarity or autocorrelation driven by an auxiliary variable, and the augmented transition function is bijective.

Next, we introduce our theoretically-informed learning algorithm based on variational auto-encoders, where we build the identifiability conditions into the space of prior distributions of the latent states and the process noise. Our framework ensures that the estimated *forward* transition function provably converges to the true unknown transition function under our identifiability assumptions.

---

[1]Here, the term "identification" encapsulates not only the estimation but also a theoretical framework showing in what circumstances the estimated model recovers the *ground-truth generative model* as in the nonlinear ICA literature (Hyvärinen et al., 2023).

Our findings show that our framework recovers latent states almost perfectly (Sec. 4.1), predicts the future states of three different dynamical systems much more accurately than state-of-the-art baselines (Secs. 4.1, 4.2 and 4.4), can be quickly adapted to new environments (Sec. 4.3), and enforcing our identifiability assumptions consistently improves identification and future prediction accuracy (App. E.2).

## 2 AN IDENTIFIABLE DYNAMICAL SYSTEM FRAMEWORK

This section starts with the notation and our generative dynamical model that leads to identifiable variables and functions under certain assumptions. In Sec. 2.2 we state the assumptions to achieve identifiability, followed by our main theoretical contribution in Sec. 2.3.

### 2.1 DATA GENERATING PROCESS

We are interested in inferring latent dynamical systems from high-dimensional observations $\mathbf{x}_{1:T}$,[2] where $\mathbf{x}_t \in \mathbb{R}^D$, with $t$ the time point. As usual, we assume that the observations are generated from a sequence of latent states $\mathbf{z}_{1:T}$, with $\mathbf{z}_t \in \mathbb{R}^K$, through an emission function $\mathbf{g} : \mathbb{R}^K \to \mathbb{R}^D$:

$$\mathbf{x}_t = \mathbf{g}(\mathbf{z}_t).$$

Without loss of generality, the latent states $\mathbf{z}_{1:T}$ evolve as a first-order Markov process[3]:

$$\mathbf{z}_t = \mathbf{f}(\mathbf{z}_{t-1}, \mathbf{s}_t),$$

where $\mathbf{f} : \mathbb{R}^{2K} \to \mathbb{R}^K$ is an auto-regressive transition function and the process noise $\mathbf{s}_t \in \mathbb{R}^K$ represents additional variables influencing the state evolution, e.g., random or unmodeled external variations in the process, not captured by deterministic dynamics. For the cartpole example, this would correspond to the strength and direction of the wind at any time.

**Challenges in identifying the latent transition function.** The transition function $\mathbf{f}(\mathbf{z}_{t-1}, \mathbf{s}_t) = \mathbf{z}_t$ maps a higher-dimensional space $\mathbb{R}^{2K}$ to a lower-dimensional space $\mathbb{R}^K$, rendering it inherently non-bijective. Consequently, multiple transition functions can correspond to the same sequence of latent variables, resulting in non-identifiability.

**Augmented dynamics.** We solve this challenge by augmenting the transition function $\mathbf{f}$ and the emission function $\mathbf{g}$ with previous time steps, resulting in the following generative process:

$$\mathbf{z}_0 \sim p_{\mathbf{z}_0}(\mathbf{z}_0), \qquad\qquad\qquad\qquad\qquad \texttt{\# initial state} \qquad (1)$$

$$\mathbf{s}_t \sim p_{\mathbf{s}|\mathbf{u}}(\mathbf{s}_t|\mathbf{u}) = \prod_k p_{s_k|\mathbf{u}}(s_{kt}|\mathbf{u}), \qquad \forall t \in 1,\ldots,T, \quad \texttt{\# process noise} \qquad (2)$$

$$\begin{bmatrix} \mathbf{z}_t \\ \mathbf{z}_{t-1} \end{bmatrix} = \mathbf{f}_{\text{aug}}\left(\begin{bmatrix} \mathbf{s}_t \\ \mathbf{z}_{t-1} \end{bmatrix}\right) = \begin{bmatrix} \mathbf{f}(\mathbf{z}_{t-1}, \mathbf{s}_t) \\ \mathbf{z}_{t-1} \end{bmatrix}, \quad \forall t \in 1,\ldots,T, \quad \texttt{\# state transition} \qquad (3)$$

$$\begin{bmatrix} \mathbf{x}_t \\ \mathbf{x}_{t-1} \end{bmatrix} = \mathbf{g}_{\text{aug}}\left(\begin{bmatrix} \mathbf{z}_t \\ \mathbf{z}_{t-1} \end{bmatrix}\right) = \begin{bmatrix} \mathbf{g}(\mathbf{z}_t) \\ \mathbf{g}(\mathbf{z}_{t-1}) \end{bmatrix}, \qquad \forall t \in 2,\ldots,T, \quad \texttt{\# observation} \qquad (4)$$

where $\mathbf{u}$ is an auxiliary variable modulating the noise distribution $p_{\mathbf{s}|\mathbf{u}}$ that implies a non-i.i.d. structure in the data generating process. For the auxiliary variable $\mathbf{u}$, we consider two practical use cases:

- Setting $\mathbf{u}$ to an *observed* regime label $r = 1,\ldots,R$ leads to a **nonstationary process noise** similar to Hyvärinen & Morioka (2016), where the data is observed under $R$ distinct regimes. For the cartpole system, this would imply several observation regimes under different wind conditions, for which their distributions of process noise are sufficiently different.

- Setting $\mathbf{u}$ to the *unobserved* noise $\mathbf{s}_{t-1}$ at previous time step implies an **autocorrelated noise process** similar to Hyvärinen & Morioka (2017). For the cartpole system, this would imply a single, windy environment where the wind speed or direction changes continuously.

---

[2]In practice, we observe multiple sequences, but we drop the sequence index in all notation for clarity. The model definition trivially generalizes to multiple sequences under exchangeability.

[3]Higher-order processes can be represented as first-order ones by state augmentation.

We build on Morioka et al. (2021), whose approach suffers from two key shortcomings: First, their approach does not learn the transitions $\mathbf{f}$. Second and more importantly, their theory is constrained to autoregressive *transitions between pairs of observational data* $(\mathbf{x}_{t-1}, \mathbf{x}_t)$. This becomes impractical when modeling high-dimensional raw signals, such as videos, which requires modeling *latent dynamics* (Hafner et al., 2019b). To tackle these, we propose a new latent dynamical framework in which we apply the augmentation within the latent space, rather than in the observational space. This requires augmenting the compositional function $\mathbf{f} \circ \mathbf{g}$ rather than just a single function $\mathbf{f}$, complicating the analysis. Finally, identifiability of the latent transition function follows when assuming that the *augmented* dynamics function is bijective (in addition to other identifiability assumptions in Sec. 2.2).

## 2.2 IDENTIFIABILITY THEORY

Let $\mathcal{M} = (\mathbf{f}_{\text{aug}}, \mathbf{g}_{\text{aug}}, p_{\mathbf{s}|\mathbf{u}})$ denote the ground-truth model. By fitting the observed sequences, we learn a model $\hat{\mathcal{M}} = (\hat{\mathbf{f}}_{\text{aug}}, \hat{\mathbf{g}}_{\text{aug}}, \hat{p}_{\mathbf{s}|\mathbf{u}})$ such that the ground-truth and the learned densities are observationally equivalent: $p_{\mathbf{f}_{\text{aug}}, \mathbf{g}_{\text{aug}}, p_{\mathbf{s}|\mathbf{u}}}(\{\mathbf{x}_t\}_{t=1}^T) = p_{\hat{\mathbf{f}}_{\text{aug}}, \hat{\mathbf{g}}_{\text{aug}}, \hat{p}_{\mathbf{s}|\mathbf{u}}}(\{\mathbf{x}_t\}_{t=1}^T), \forall \mathbf{x}_t \in \mathcal{X}$ for $t \in 1, \ldots, T$. We make the following assumptions:

*(A1)* **Injectivity and bijectivity.** The emission functions $\mathbf{g}$ and $\hat{\mathbf{g}}$ are injective, implying the same for the augmented functions $\mathbf{g}_{\text{aug}}, \hat{\mathbf{g}}_{\text{aug}}$. The augmented transition functions $\mathbf{f}_{\text{aug}}, \hat{\mathbf{f}}_{\text{aug}}$ are bijective.

*(A2)* **Conditionally independent noise.** A single latent variable $z_{kt}$ is influenced by a single process noise variable $s_{kt}$: $z_{kt} = f_k(\mathbf{z}_{t-1}, s_{kt})$. Let $q_k(s_{kt}, \mathbf{u}) = \log p(s_{kt}|\mathbf{u})$ denote the conditional log-density of the noise variable $s_{kt}$. Let $\eta_k(z_{kt}, \mathbf{u}) = \log p(z_{kt}|\mathbf{z}_{t-1}, \mathbf{u})$ denote the conditional log-density of the state variable $z_{kt}$. Conditioned on the auxiliary variable $\mathbf{u}$, the log densities decompose for all $t \in 1, \ldots, T$:

$$\log p(\mathbf{s}_t|\mathbf{u}) = \sum_{k=1}^K \underbrace{\log p(s_{kt}|\mathbf{u})}_{q_k(s_{kt}, \mathbf{u})} = \sum_{k=1}^K q_k(s_{kt}, \mathbf{u}), \tag{5}$$

$$\log p(\mathbf{z}_t|\mathbf{z}_{t-1}, \mathbf{u}) = \sum_{k=1}^K \underbrace{\log p(z_{kt}|\mathbf{z}_{t-1}, \mathbf{u})}_{\eta_k(z_{kt}, \mathbf{u})} = \sum_{k=1}^K \eta_k(z_{kt}, \mathbf{u}). \tag{6}$$

*(A3)* **Sufficient variability of latent state** $\mathbf{z}_t$**.** For any $\mathbf{z}_t$, there exist some $2K$ values of $\mathbf{u}$: $\mathbf{u}_1, \ldots, \mathbf{u}_{2K}$, such that the $2K$ vectors

$$\mathbf{v}_l(\mathbf{z}_t, \mathbf{u}_1), \ldots, \mathbf{v}_l(\mathbf{z}_t, \mathbf{u}_{2K}) \tag{7}$$

are linearly independent for some index $l$ of the auxiliary variable $\mathbf{u}$, where

$$\mathbf{v}_l(\mathbf{z}_t, \mathbf{u}) = \left( \frac{\partial^2 \eta_1(z_{1t}, \mathbf{u})}{\partial z_{1t} \partial u_l}, \cdots, \frac{\partial^2 \eta_K(z_{Kt}, \mathbf{u})}{\partial z_{Kt} \partial u_l}, \frac{\partial^3 \eta_1(z_{1t}, \mathbf{u})}{\partial z_{1t}^2 \partial u_l}, \cdots, \frac{\partial^3 \eta_K(z_{Kt}, \mathbf{u})}{\partial z_{Kt}^2 \partial u_l} \right). \tag{8}$$

*(A4)* **Sufficient variability of process noise** $\mathbf{s}_t$**.** For any $\mathbf{s}_t$, there exist some $2K$ values of $\mathbf{u}$: $\mathbf{u}_1, \ldots, \mathbf{u}_{2K}$, such that the $2K$ vectors

$$\mathbf{w}_l(\mathbf{s}_t, \mathbf{u}_1), \ldots, \mathbf{w}_l(\mathbf{s}_t, \mathbf{u}_{2K}) \tag{9}$$

are linearly independent for some index $l$ of the auxiliary variable $\mathbf{u}$, where

$$\mathbf{w}_l(\mathbf{s}_t, \mathbf{u}) = \left( \frac{\partial^2 q_1(s_{1t}, \mathbf{u})}{\partial s_{1t} \partial u_l}, \cdots, \frac{\partial^2 q_K(s_{Kt}, \mathbf{u})}{\partial s_{Kt} \partial u_l}, \frac{\partial^3 q_1(s_{1t}, \mathbf{u})}{\partial s_{1t}^2 \partial u_l}, \cdots, \frac{\partial^3 q_K(s_{Kt}, \mathbf{u})}{\partial s_{Kt}^2 \partial u_l} \right). \tag{10}$$

**Discussion of Assumptions.** In Assumption *(A1)*, the bijectivity of the augmented latent transition function captures the functional dependence between a *latent pair* $(\mathbf{z}_{t-1}, \mathbf{z}_t)$ and the process noise $\mathbf{s}_t$. This differs from Morioka et al. (2021), who use an augmented transition model on *observations* $(\mathbf{x}_{t-1}, \mathbf{x}_t)$. Returning to the cartpole example, the transition function $\mathbf{f}$ would map a latent state $\mathbf{z}_t$ to two different future states $\mathbf{z}_{t+1}^{(1)}, \mathbf{z}_{t+1}^{(2)}$ if the corresponding wind variables $\mathbf{s}_{t+1}^{(1)}, \mathbf{s}_{t+1}^{(2)}$ differ. This assumption would not hold if the controller or the transition function has intrinsic stochasticity, breaking the bijectivity. Assumptions *(A3, A4)* imply that the latent states and the process noise are sufficiently

different across different environments, similar to the autocorrelation and non-stationarity conditions in Hyvärinen et al. (2019; 2023). Assumption *(A3)* generalizes the sufficient variability condition defined for a regime label in Yao et al. (2021), to a general auxiliary variable $\mathbf{u}$. For *(A3)* to hold, the changes in the wind distribution should sufficiently change the distribution of latent states, i.e., 2D positions and velocities. For *(A4)* to hold, we need to observe the cartpole system in sufficiently many environments with sufficiently different wind (noise) conditions. These assumptions are violated, e.g., if the wind is constant across the observed environments. For more details, see App. C.

## 2.3 MAIN THEORETICAL CONTRIBUTION

In this section, we state our main theoretical contribution, that is, the novel identifiability result for a latent transition function $\mathbf{f}$ (**Theorem 2**). We start with a theorem on the identifiability of the conditionally independent latent states $\mathbf{z}_t | \mathbf{z}_{t-1}, \mathbf{u}$ (**Theorem 1**), which is an extension of the identifiability result established for the nonstationary noise case in Yao et al. (2021). The proofs are detailed in Apps. C.1 and C.2.

**Theorem 1 (Identifiability of latent states $\mathbf{z}_{1:T}$, based on Yao et al. (2021)).** *Under assumptions (A1, A2, A3), latent states $\mathbf{z}_t$ are identifiable up to a permutation and element-wise invertible transformation, i.e., there exists a function $\mathbf{h} : \mathbb{R}^K \to \mathbb{R}^K$, such that $\mathbf{z}_t = \mathbf{h}(\hat{\mathbf{z}}_t)$, where $\mathbf{h} = \pi_z \circ r_z$ is a composition of a permutation $\pi_z : [K] \to [K]$ and an element-wise invertible transformation $r_z : \mathbb{R}^K \to \mathbb{R}^K$. Equivalently, the same holds for the emission function, i.e., $\hat{\mathbf{g}} = \mathbf{g} \circ \mathbf{h}$.*

**Theorem 2 (Identifiability of the transition function f).** *Under assumptions (A1, A2, A3, A4), the process noise $\mathbf{s}_t$ is identifiable up to a permutation and element-wise invertible transformation, i.e., there exists a function $\mathbf{k} : \mathbb{R}^K \to \mathbb{R}^K$, such that $\mathbf{s}_t = \mathbf{k}(\hat{\mathbf{s}}_t)$, where $\mathbf{k} = \pi_s \circ r_s$ is a composition of a permutation $\pi_s : [K] \to [K]$ and an element-wise invertible transformation $r_s : \mathbb{R}^K \to \mathbb{R}^K$. Equivalently, the same holds for the transition function $\hat{\mathbf{f}}_{aug} = \mathbf{h}_{aug}^{-1} \circ \mathbf{f}_{aug} \circ \mathbf{k}_{aug}$, since augmented functions $\mathbf{h}_{aug}^{-1}$ and $\mathbf{k}_{aug}$ decompose into block-wise functions: $\mathbf{h}_{aug}^{-1} = [\mathbf{h}^{-1}, \mathbf{h}^{-1}]$ and $\mathbf{k}_{aug} = [\mathbf{k}, \mathbf{h}]$, such that $\mathbf{k}$ and $\mathbf{h}$ are already shown to be compositions of a permutation and an element-wise invertible transformation.*

**Proof sketch.** Latent states $\mathbf{z}_t$ and $\mathbf{z}_{t-1}$ are identifiable under sufficient variability and conditional independence assumptions on states $\mathbf{z}$. Intuitively, the idea is also to identify the noise variables $\mathbf{s}_t$, which leads to the identifiability of the transition function as $\mathbf{z}_t = \mathbf{f}(\mathbf{z}_{t-1}, \mathbf{s}_t)$.

In the proof in App. C.2, we first show that the function $\mathbf{k}_{\text{aug}} : \mathbb{R}^{2K} \to \mathbb{R}^{2K}$ that maps the learned pair $(\hat{\mathbf{s}}_t, \hat{\mathbf{z}}_{t-1})$ to the ground-truth pair $(\mathbf{s}_t, \mathbf{z}_{t-1})$, $\begin{bmatrix} \mathbf{s}_t \\ \mathbf{z}_{t-1} \end{bmatrix} = \mathbf{k}_{\text{aug}}\left( \begin{bmatrix} \hat{\mathbf{s}}_t \\ \hat{\mathbf{z}}_{t-1} \end{bmatrix} \right) = \begin{bmatrix} \mathbf{k}_1(\hat{\mathbf{s}}_t, \hat{\mathbf{z}}_{t-1}) \\ \mathbf{k}_2(\hat{\mathbf{s}}_t, \hat{\mathbf{z}}_{t-1}) \end{bmatrix}$, is bijective. Next, we show that the function components $\mathbf{k}_1, \mathbf{k}_2 : \mathbb{R}^{2K} \to \mathbb{R}^K$ that take the pair $(\hat{\mathbf{s}}_t, \hat{\mathbf{z}}_{t-1})$ as input are equivalent to invertible element-wise functions $\mathbf{k}, \mathbf{h} : \mathbb{R}^K \to \mathbb{R}^K$, such that (i) $\mathbf{k}$ only depends on $\hat{\mathbf{s}}_t$, i.e., $\mathbf{s}_t = \mathbf{k}_1(\hat{\mathbf{s}}_t, \hat{\mathbf{z}}_t) = \mathbf{k}(\hat{\mathbf{s}}_t)$ and (ii) $\mathbf{h}$ only depends on $\hat{\mathbf{z}}_{t-1}$, i.e., $\mathbf{z}_{t-1} = \mathbf{k}_2(\hat{\mathbf{s}}_t, \hat{\mathbf{z}}_{t-1}) = \mathbf{h}(\hat{\mathbf{z}}_{t-1})$. For (ii), we use the result of **Theorem 1**. For (i), we use the assumption *(A4)*, the sufficient variability in the process noise.

## 3 PRACTICAL IMPLEMENTATION USING VARIATIONAL INFERENCE

We turn our theoretical framework into a practical implementation using variational inference, which approximates the true posterior over the noise and the latent states given the observations. In contrast to previous works that approximate the *inverse* transition function (Yao et al., 2021; 2022), we *directly estimate the forward transition function*. To achieve this, we first map the observation $\mathbf{x_t}$ to process noise $\mathbf{s}_t$, from which we then obtain the latent states $\mathbf{z}_t$ similar to Franceschi et al. (2020). Different from Franceschi et al. (2020), our process noise prior $p_\theta(\mathbf{s}|\mathbf{u})$ does not depend on the previous state $\mathbf{z}_{t-1}$, as this dependence would make the model non-identifiable. In the implementation and the experiments, we set the auxiliary variable $\mathbf{u}$ to a regime label where the data is observed under distinct regimes.

For a theory-informed implementation, we build our assumptions into the model as follows. We enforce (i) conditionally independent process noise by using independent 1D conditional flows for

$p_\theta(s_{kt}|\mathbf{u})$, and (ii) conditionally independent latent states by modeling each output $k$ of the transition function $f_k(\mathbf{z}_{t-1}, s_{kt})$ as a separate MLP. We provide the learning algorithm in **Algorithm 1**. For space considerations, we provide the implementation details in Fig. 6 and Apps. D and F.3.

**Sequential prediction of $\mathbf{s}_{1:T}$ and $\mathbf{z}_{0:T}$.** Each observation $\mathbf{x}_t$ is mapped to an intermediate embedding $\tilde{\mathbf{x}}_t$ via an MLP or CNN backbone depending on the input modality. To infer the initial latent state, an encoder (MLP) takes the embeddings $\tilde{\mathbf{x}}_{1:T_{\mathrm{ic}}}$ up to time $T_{\mathrm{ic}}$ as input. It outputs the parameters of the initial state posterior, $q_\phi(\mathbf{z}_0|\tilde{\mathbf{x}}_{1:T_{\mathrm{ic}}})$, from which the initial state $\mathbf{z}_0$ is sampled: $\mathbf{z}_0 \sim q_\phi(\mathbf{z}_0|\tilde{\mathbf{x}}_{1:T_{\mathrm{ic}}})$.

Once the initial state is sampled, the model predicts the noise $\mathbf{s}_t$ and the latent state $\mathbf{z}_t$ iteratively for each time step $t = 1, \ldots, T$. At each step, the embeddings up to time $t$, $\tilde{\mathbf{x}}_{1:t}$, along with the previously sampled state $\mathbf{z}_{t-1}$, are used to infer the process noise. A forward sequential model (RNN + MLP) takes $[\tilde{\mathbf{x}}_{1:t}, \mathbf{z}_{t-1}]$ as input to compute the parameters of the noise posterior, $q_\phi(\mathbf{s}_t|\tilde{\mathbf{x}}_{1:t}, \mathbf{z}_{t-1})$, from which the noise is sampled: $\mathbf{s}_t \sim q_\phi(\mathbf{s}_t|\tilde{\mathbf{x}}_{1:t}, \mathbf{z}_{t-1})$. Using the sampled noise $\mathbf{s}_t$ and the previous state $\mathbf{z}_{t-1}$, the next state $\mathbf{z}_t$ is predicted through the transition function $\mathbf{z}_t = \mathbf{f}(\mathbf{z}_{t-1}, \mathbf{s}_t)$. Specifically, the components of $\mathbf{z}_t$ are computed as $z_{kt} = f_k(\mathbf{z}_{t-1}, s_{kt})$, for $k = 1, \ldots, K$. Due to the sequential computation, the time complexity of a forward-pass is linear in the input sequence length $\mathcal{O}(T)$.

**ELBO computation and priors.** The decoder takes the latent state $\mathbf{z}_t$ as input and outputs the parameters of the observation likelihood, $p_\theta(\mathbf{x}_t|\mathbf{d}(\mathbf{z}_t))$. For the ELBO computation, we weigh the KL term with the hyperparameter $\beta$ (Higgins et al., 2017), which is selected through validation. We assume a standard Gaussian prior for the initial state, $p(\mathbf{z}_0) = \mathcal{N}(0, I)$. We use the auxiliary variable $\mathbf{u}$ in implementing the prior for the noise variables $\mathbf{s}_t$, $p_\theta(\mathbf{s}_t|\mathbf{u}) = \prod_k p_\theta(s_{kt}|\mathbf{u})$, where each $p_\theta(s_{kt}|\mathbf{u})$ is modeled as an independent, trainable 1D conditional flow. To allow for multi-modal prior distributions of the 1D noise variables $s_{kt}$, we employ neural spline flows (Durkan et al., 2019).

---

**Algorithm 1** Practical learning algorithm

---

**Requires:** Variational posterior networks (`ICEncoder` and `NoiseEncoder`) and `Decoder`

1. Encode initial condition parameters: $\mu_{\mathbf{z}_0}, \log \sigma_{\mathbf{z}_0}^2 = \texttt{ICEncoder}(\mathbf{x}_{1:T_{\mathrm{ic}}})$

2. Sample initial condition: $\mathbf{z}_0 \sim \mathcal{N}(\mu_{\mathbf{z}_0}, \sigma_{\mathbf{z}_0}^2 \mathbf{I})$

3. For $t \in 1, \ldots, T$:

   (a) Encode noise parameters: $\mu_{\mathbf{s}_t}, \log \sigma_{\mathbf{s}_t}^2 = \texttt{NoiseEncoder}(\mathbf{x}_{1:t}, \mathbf{z}_{t-1})$

   (b) Sample noise: $\mathbf{s}_t \sim \mathcal{N}(\mu_{\mathbf{s}_t}, \sigma_{\mathbf{s}_t}^2 \mathbf{I})$

   (c) Compute the next latent state: $\mathbf{z}_t = \mathbf{f}(\mathbf{s}_t, \mathbf{z}_{t-1})$

   (d) Decode: $\mathbf{x}_t = \texttt{Decoder}(\mathbf{z}_t)$

4. Compute ELBO: $\mathcal{L} = \mathcal{L}_{\mathrm{R}} - \beta \mathcal{L}_{\mathrm{KL}}$. Samples $\{\mathbf{s}_{1:T}, \mathbf{z}_{0:T}\}$ are used to approximate $\mathcal{L}_{\mathrm{KL}}$:

$$\mathcal{L}_{\mathrm{R}} = \sum\nolimits_{t=1}^{T} \mathbb{E}_{q_\phi(\mathbf{z}_t|\ldots)}[\log p_\theta(\mathbf{x}_t|\mathbf{z}_t)],$$
$$\mathcal{L}_{\mathrm{KL}} = D_{\mathrm{KL}}(q_\phi(\mathbf{z}_0|\mathbf{x}_{1:T})\|p_\theta(\mathbf{z}_0)) + \sum\nolimits_{t=1}^{T} \mathbb{E}_{q_\phi(\mathbf{z}_{t-1}|\ldots)}[D_{\mathrm{KL}}(q_\phi(\mathbf{s}_t|\mathbf{z}_{t-1}, \mathbf{x}_{1:t})\|p_\theta(\mathbf{s}_t|\mathbf{u}))].$$

5. Update the parameters $\{\theta, \phi\}$.

---

## 4 EXPERIMENTS

In this section, we evaluate our method's ability to *(i)* recover true system dynamics in controlled setups to validate our theory (Sec. 4.1), *(ii)* accurately forecast long horizons in complex synthetic and real-world datasets (Secs. 4.1 and 4.2), and *(iii)* adapt to unseen environments efficiently (Sec. 4.3). In addition, we compare joint vs. two-stage dynamics training in Sec. 4.4, present an ablation study and additional figures in App. E, and further details in App. F. Our implementation to reproduce the study can be found at `https://github.com/caglar-hizli/idf-latent-dyn`.

**Baselines.** For latent dynamics identification, we compare with state-of-the-art identifiable representation learning (IRL) methods: $\beta$-VAE (Higgins et al., 2017), LEAP (linear: LEAP-LIN, non-parametric: LEAP-NP; Yao et al., 2021), and TDRL (Yao et al., 2022). For sample-efficient adaptation, we compare with the best-performing IRL method, LEAP-LIN. For future prediction, we compare with state-of-the-art neural state-space, ODE and SDE methods: KALMANVAE (Fraccaro et al., 2017), CRU (Schirmer et al., 2022), NODE (Chen et al., 2018), Latent NODE (Rubanova et al., 2019), ODE$^2$VAE (Yildiz et al., 2019), MONODE (Auzina et al., 2024), and Latent SDE (Li et al., 2020).

Table 1: Results for synthetic and cartpole datasets (mean $\pm$ std.dev. across 5 seeds). Rows are marked (N/A) for methods unable to predict the future or the process noise. LEAP-NP and TDRL cannot generate future predictions as they only approximate the inverse transition function. MCC: mean correlation coefficient; higher is better $\uparrow$. $\text{MSE}[\bar{\mathbf{x}}_{\text{future}[\tau]}]$ denotes mean squared error over $\tau$ future time points; lower is better $\downarrow$. Best **bolded** based on Welch's $t$-test with $p < 0.01$.

| DATASET | METRICS | MODELS | | | | | | |
|---|---|---|---|---|---|---|---|---|
| | | $\beta$-VAE | KALMANVAE | CRU | LEAP-LIN | LEAP-NP | TDRL | OURS |
| SYNTH. | $\text{MCC}[\bar{\mathbf{z}}_{\text{input}}] \uparrow$ | $0.60_{\pm0.05}$ | $0.64_{\pm0.05}$ | $0.47_{\pm0.05}$ | $0.68_{\pm0.03}$ | $\mathbf{0.89}_{\pm0.04}$ | $0.73_{\pm0.04}$ | $\mathbf{0.94}_{\pm0.09}$ |
| | $\text{MCC}[\bar{\mathbf{s}}_{\text{input}}] \uparrow$ | N/A | N/A | N/A | $0.14_{\pm0.01}$ | $0.26_{\pm0.04}$ | $0.42_{\pm0.11}$ | $\mathbf{0.64}_{\pm0.13}$ |
| | $\text{MSE}[\bar{\mathbf{x}}_{\text{future}[2]}] \downarrow$ | N/A | $1.27_{\pm0.19}$ | $8.85_{\pm0.82}$ | $0.22_{\pm0.03}$ | N/A | N/A | $\mathbf{0.06}_{\pm0.02}$ |
| | $\text{MSE}[\bar{\mathbf{x}}_{\text{future}[4]}] \downarrow$ | N/A | $1.32_{\pm0.27}$ | $8.75_{\pm0.85}$ | $0.18_{\pm0.03}$ | N/A | N/A | $\mathbf{0.08}_{\pm0.02}$ |
| | $\text{MSE}[\bar{\mathbf{x}}_{\text{future}[8]}] \downarrow$ | N/A | $1.72_{\pm0.82}$ | $9.08_{\pm0.94}$ | $0.59_{\pm0.13}$ | N/A | N/A | $\mathbf{0.20}_{\pm0.08}$ |
| CARTPOLE | $\text{MCC}[\bar{\mathbf{z}}_{\text{input}}] \uparrow$ | $0.68_{\pm0.01}$ | $0.65_{\pm0.07}$ | $0.85_{\pm0.05}$ | $0.81_{\pm0.07}$ | $0.76_{\pm0.07}$ | $0.56_{\pm0.09}$ | $\mathbf{0.95}_{\pm0.02}$ |
| | $\text{MCC}[\bar{\mathbf{z}}_{\text{future}[8]}] \uparrow$ | N/A | $0.73_{\pm0.05}$ | $0.70_{\pm0.06}$ | $0.78_{\pm0.05}$ | N/A | N/A | $\mathbf{0.91}_{\pm0.02}$ |

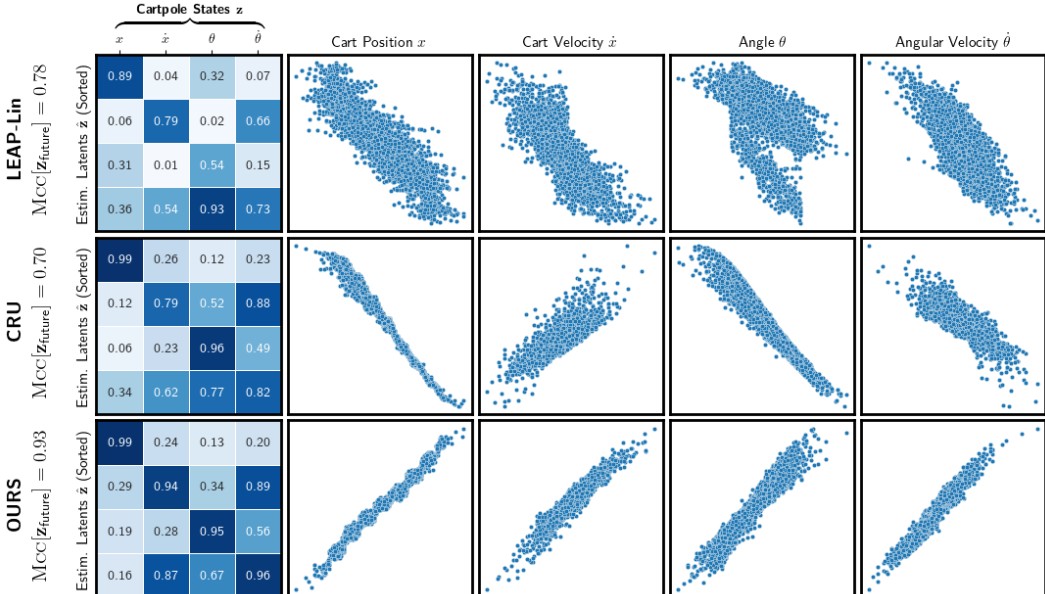

Figure 2: Correlation plots for first 8 input points. For illustration purposes, we use the data from the seed with median performance. Cartpole system has 4 true states: cart position $x$ and velocity $\dot{x}$, pole angle $\theta$ and angular velocity $\dot{\theta}$. **Left column:** The correlation matrix between ground-truth states and latent estimations for best-performing 3 models. Columns correspond to ground-truth states $x, \dot{x}, \theta, \dot{\theta}$, while rows correspond to most highly-correlated estimated latent dimensions. Darker colors on diagonal imply better one-to-one mapping. **Next 4 columns:** 4 scatter plots between each ground-truth state dimension and its aligned latent. Here, a diagonal line implies better one-to-one mapping. An extended version, also covering the last $T_{\text{future}}$ time steps, is available in Fig. 10 in App. E.

**Metrics.** For latent dynamics identification, we use mean correlation coefficient (MCC) between the ground-truth states and the estimated latents, which is the standard metric in the nonlinear ICA literature (Hyvärinen et al., 2019). $\text{MCC}[\bar{\mathbf{z}}_{\text{input}}]$ and $\text{MCC}[\bar{\mathbf{s}}_{\text{input}}]$ denote MCC on latent states and process noise for the first $T_{\text{input}}$ steps. For future prediction, we use $\text{MCC}[\bar{\mathbf{z}}_{\text{future}}]$ on future latent states and the mean squared error $\text{MSE}[\bar{\mathbf{x}}_{\text{future}}]$ on the future observations $\bar{\mathbf{x}}_{\text{future}}$.

**Datasets.** In all experiments, the data is observed under $R$ distinct regimes, and we set the auxiliary variable $\mathbf{u}$ to the one-hot encoded regime label $r$. *Synthetic:* To validate our theory, we generate a synthetic dataset satisfying our identifiability assumptions as commonly done in theoretical works on IRL. We follow the setups in Yao et al. (2021; 2022). We have $R = 20$ regimes with nonstationary noise: the (conditional) distribution of noise $\mathbf{s}$ varies between regimes. To evaluate the long-horizon forecast performance, we use $T_{\text{dyn}} = 4$ observations for training the dynamical model and $T_{\text{future}} = 8$

observations for evaluating future predictions. *Cartpole:* We use the setup as described in Yao et al. (2022), with one modification: Yao et al. only use fixed action sequences, i.e., only always go-right or always go-left, which is not representative of real-world scenarios. Instead, we implement continuous actions to obtain a more realistic setup. We have $R = 6$ regimes (5 for training, 1 for testing), with distinct gravity values. The underlying dynamics are deterministic; the noise $\mathbf{s}$ accounts for the variability due to unknown initial conditions and different gravity values between the domains. *Mocap:* To evaluate future predictions on long horizons with complex real-world dynamics, we use three CMU motion capture (Mocap) datasets from the dynamical systems literature. The different regimes correspond to different persons walking. MOCAP-SINGLE (Yildiz et al., 2019; Li et al., 2020) contains walking sequences of a single subject ($R = 1$), and MOCAP-MULTI and MOCAP-SHIFT (Auzina et al., 2024) consist of sequences of 6 different subjects ($R = 6$); in MOCAP-SHIFT, one subject is left out for testing. Here, we interpret the process noise $\mathbf{s}$ as modeling the external variations in the observed data, which might show nonstationary behavior due to subject-specific characteristics such as limb lengths, joint angles, sensor positioning, sensor noise, or initial conditions.

## 4.1 OUR MODEL RECOVERS TRUE SYSTEM DYNAMICS BETTER IN CONTROLLED SETUPS

We present MCC and MSE results for the synthetic and cartpole datasets in Table 1. In the synthetic dataset, all IRL methods (LEAP-LIN, LEAP-NP (Yao et al., 2021); TDRL (Yao et al., 2022); and OURS) estimate latent states with high correlation to the true states, as indicated by $\text{MCC}[\bar{\mathbf{z}}_{\text{input}}]$. Our method has the highest $\text{MCC}[\bar{\mathbf{z}}_{\text{input}}]$ and $\text{MCC}[\bar{\mathbf{s}}_{\text{input}}]$ scores among all methods.

High values of $\text{MCC}[\bar{\mathbf{z}}_{\text{input}}]$ and $\text{MCC}[\bar{\mathbf{s}}_{\text{input}}]$ indicate a close alignment between the estimated and true latent transition functions, as stated in **Theorem 2**. This is validated by our method's superior future prediction accuracy, with lower $\text{MSE}[\bar{\mathbf{x}}_{\text{future}}]$ across 2-, 4-, and 8-step horizons. It is important to note that the IRL methods LEAP-NP and TDRL cannot generate future predictions (N/A) since they approximate only the inverse transition function. As the prediction horizon increases—e.g., to 8 steps, twice as much as observed during training—the performance gap between our method and the baselines widens further. See predicted trajectories with calibrated uncertainties in Fig. 11 in App. E.

For cartpole, methods that model complex temporal dependencies (LEAP-LIN, LEAP-NP (Yao et al., 2021); CRU (Schirmer et al., 2022); and OURS) outperform others in $\text{MCC}[\bar{\mathbf{z}}_{\text{input}}]$ and $\text{MCC}[\bar{\mathbf{z}}_{\text{future}}]$. For the top-performing models (LEAP-LIN (Yao et al., 2021), CRU (Schirmer et al., 2022), and OURS), we present MCC correlation matrices and scatter plots comparing the ground-truth states with the 4 most correlated latent dimensions for the first $T_{\text{input}}$ time steps in Fig. 2. An extended version of this figure, also covering the last $T_{\text{future}}$ time steps, is available in Fig. 10 in App. E. LEAP-LIN (Yao et al., 2021) struggles to estimate well-aligned latents for both the first 8 input steps and the subsequent 8 future steps. CRU (Schirmer et al., 2022) recovers well-aligned latents for cart position $x$ and angle $\theta$ during the first 8 steps but loses accuracy in future predictions. In contrast, our method consistently recovers the most highly correlated latents, with the performance gap increasing for future predictions.

**Summary.** Our method recovers the true latent dynamics more accurately than baselines in both synthetic and cartpole setups, as shown by (i) higher MCC for input latents and (ii) more accurate future predictions, with the performance gap widening as the prediction horizon increases.

## 4.2 OUR MODEL PREDICTS LONG-HORIZON REAL-WORLD DYNAMICS MORE ACCURATELY

We present the MSE results in Tab. 2 for all Mocap datasets. These datasets exhibit complex real-world dynamics with prediction horizons of 75-300 steps. Across all datasets, NODE (Chen et al., 2018) performs the worst, highlighting the need to model latent dynamics. As an additional strong baseline, we trained CRU (Schirmer et al., 2022) on Mocap datasets. Although this approach based on Kalman filters fits the training data well, it reverts to its linear prior when extrapolating into

Table 2: TEST-MSE($\downarrow$) for MOCAP-SINGLE, MOCAP-MULTI and MOCAP-SHIFT. [†]Results from Li et al. (2020). [††]Results from Auzina et al. (2024).

| MODELS | MOCAP-SINGLE | MOCAP-MULTI | MOCAP-SHIFT |
|---|---|---|---|
| LEAP-LIN | $17.08_{\pm 2.24}$ | $57.9_{\pm 10.5}$ | $51.1_{\pm 2.8}$ |
| NODE | $22.49_{\pm 0.88}$[†] | $72.2_{\pm 12.4}$[††] | $61.6_{\pm 6.2}$[††] |
| ODE$^2$VAE | $8.09_{\pm 1.95}$[†] | - | - |
| Latent NODE | $5.98_{\pm 0.28}$[†] | - | - |
| Latent SDE | $4.03_{\pm 0.20}$[†] | - | - |
| MONODE | - | $57.7_{\pm 9.8}$[††] | $58.0_{\pm 10.7}$[††] |
| OURS | $\mathbf{3.82}_{\pm 0.40}$ | $\mathbf{19.6}_{\pm 1.2}$ | $\mathbf{36.0}_{\pm 1.9}$ |

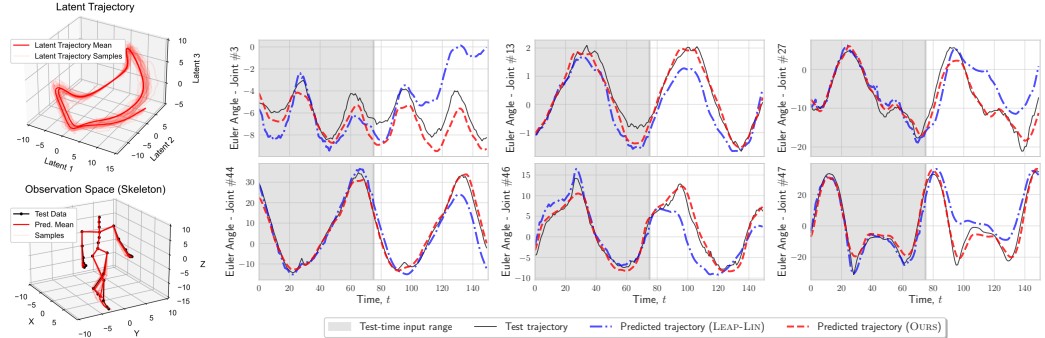

Figure 3: **Left:** Latent trajectories and skeletons for OURS model in MOCAP-SINGLE (observation trajectories in Fig. 8). **Right:** Test trajectory predictions for LEAP-LIN and OURS in MOCAP-MULTI. At test time, first 75 time points are given as input, and the full prediction horizon is $T = 150$ steps.

the (unseen) future. Consequently, it fails to predict the unseen time points, and we have omitted its results from this section.

MOCAP-SINGLE consists of sequences from a single subject, which violates our identifiability assumptions. Despite this model misspecification, our method achieves the lowest MSE. In contrast, MOCAP-MULTI and MOCAP-SHIFT show non-i.i.d. structures, with multiple subjects treated as distinct environments. Our method is particularly well-suited for these datasets, significantly outperforming other baselines, including MONODE (Auzina et al., 2024), which is designed to handle non-i.i.d. structures by predicting environment-specific dynamic variables.

In Fig. 3 **(Right)**, we present the predictions of our model alongside those of LEAP-LIN for six dimensions within the data space that are representative of complex dynamics (all other dimensions are included in Fig. 12 in App. E). Both models exhibit high accuracy for the initial half of the sequence, provided as input. However, our model demonstrates significantly improved extrapolation into the future, indicative of a superior approximation of the underlying system. In Fig. 3 **(Left)**, we show estimated latent trajectories, along with the generated skeletons for time $T = 300$ on MOCAP-SINGLE. The latent trajectories exhibit smooth, cyclic patterns with minimal differences between trials, reflecting the cyclic nature of walking. The estimated trajectories for the joint angles closely match the actual data trajectories, with calibrated uncertainties.

**Summary.** Our model achieves state-of-the-art performance across all datasets, outperforming strong neural ODE/SDE baselines that were specifically designed for the task. Notably, our approach performs best on MOCAP-SINGLE that violates our assumptions, suggesting that our method maintains strong real-world applicability, even in scenarios that challenge its core assumptions.

## 4.3 IDENTIFYING DYNAMICS IS THE KEY TO SUCCESSFUL ADAPTATION

In this section, we explore whether for dynamical model adaptability it is more important to identify the underlying dynamics than just the latent states. We utilize our modified cartpole dataset, consisting of five source domains with varying gravity levels $g = \{5, 10, 20, 30, 40\}$ and a target domain with $g = 90$. When tested on the target domain, the $\text{MCC}[\bar{\mathbf{z}}_{\text{future}}]$ of both LEAP-LIN (Yao et al., 2021) and our method drops significantly. To address this, we fine-tune the models on *adaptation datasets* with $N = \{20, 50, 100, 1000\}$ sequences. As a strong baseline, we also train our model on an extended dataset that includes all the data from the source and target domains.

As shown in Fig. 4, fine-tuning on just 50 sequences allows our model to recover the $\text{MCC}[\bar{\mathbf{z}}_{\text{future}}]$ performance of the strong baseline trained on the complete dataset. Fine-tuning our model on even more data makes it a better expert on the adapted do-

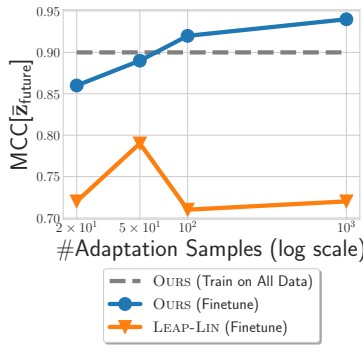

Figure 4: $\text{MCC}[\bar{\mathbf{z}}_{\text{future}}]$ with varying number of adaptation samples.

main. In contrast, the MCC score of LEAP-LIN (Yao et al., 2021) does not consistently improve, regardless of dataset size.

**Summary.** The sample-efficient adaptation performance of our method suggests that identifying dynamics plays a key role in successful dynamics adaptation, while identifying only the latent states (LEAP-LIN, Yao et al., 2021) does not guarantee successful transfer to new domains.

## 4.4 JOINTLY LEARNING LATENTS AND THEIR DYNAMICS IMPROVES FUTURE PREDICTION ACCURACY

In principle, one can use an off-the-shelf identifiable representation learning method, such as LEAP-NP (Yao et al., 2021), identify the ground-truth factors, and fit a state transition function on the inferred latent codes. In this section, we demonstrate that such a two-stage approach approximates an unknown transition function less well than our joint learning procedure that identifies the transition and emission functions. We compare the two on our synthetic dataset. To learn the unknown transitions, we train (i) a three-layer MLP (1-MLP) and (ii) $K$ three-layer MLPs ($K$-MLP), with leaky-ReLU activations on the latent state

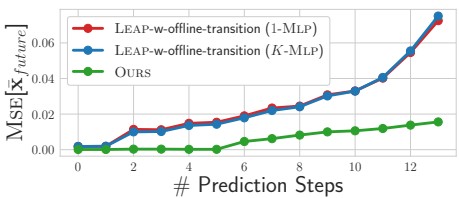

Figure 5: Comparison of MSE achieved by our approach vs. two-stage LEAP training.

sequence $\bar{\mathbf{z}}_{\text{train}}$ inferred by LEAP-NP (Yao et al., 2021). We unroll the learned transition function for an additional $T_{\text{future}}$ time steps, decode all the latent states, and compute the MSE in the data space. We repeat this three times and compare the average error with the error of our approach in Fig. 5. Our approach achieves smaller errors at all time points, and the gap widens as we predict for longer horizons. This indicates that identifying the emission and transition functions jointly improves future prediction accuracy.

## 5 DISCUSSION

We have presented the first latent dynamical system that allows for the identification of the unknown transition function, and theoretically proved its identifiability, based on standard assumptions. We evaluated our approach on synthetic data, the cartpole environment and real-world Mocap datasets, and showed that *(i)* the estimated latent states correlated strongly with the ground truth, *(ii)* our method had the highest future prediction accuracy with calibrated uncertainties, *(iii)* it could adapt to new environments using a handful of data, and *(iv)* it produces state-of-the-art future prediction results on complex real-world dynamics for long horizons, highlighting its usefulness for practical scenarios. The main limitation stems from the identifiability assumptions, which are further discussed in detail in App. C. For future work, it would be intriguing to examine how their violation might influence the final model's performance. Finally, demonstrating improved downstream performance from our method, e.g. in model-based policy learning, would be of interest.

**Acknowledgements** Çağatay Yıldız and Matthias Bethge are members of the Machine Learning Cluster of Excellence, funded by the Deutsche Forschungsgemeinschaft (DFG, German Research Foundation) under Germany's Excellence Strategy – EXC number 2064/1 – Project number 390727645. Matthia Bethge acknowledges financial support via the Open Philanthropy Foundation funded by the Good Ventures Foundation.

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

# A    RELATED WORK

Table 3: A comparison of our approach and the previous work, highlighting that our approach is the first to predict future states of a dynamical system by utilizing a transition function that identifies the underlying, true dynamics.

| | Identifies latent states | Models latent dynamics | Can predict future | Identifies latent dynamics |
|---|---|---|---|---|
| SLOWVAE, Klindt et al. (2020) | ✓ | ✗ | ✗ | ✗ |
| IIA, Morioka et al. (2021) | (✓) [†] | ✗ | ✗ | ✗ |
| LEAP, Yao et al. (2021) | ✓ | ✓ | ✗/✓ [††] | ✗ |
| TDRL, Yao et al. (2022) | ✓ | ✓ | ✗ | ✗ |
| NODE, Chen et al. (2018) | ✗ | ✗ | ✓ | ✗ |
| KALMANVAE, Fraccaro et al. (2017) Latent NODE, Rubanova et al. (2019) ODE²VAE, Yildiz et al. (2019) Latent SDE, Li et al. (2020) CRU, Schirmer et al. (2022) MONODE, Auzina et al. (2024) | ✗ | ✓ | ✓ | ✗ |
| OURS | ✓ | ✓ | ✓ | ✓ |

[†] only the process noise is identified,     [††] only for linear approximation (LEAP-LIN)

# B    IDENTIFIABILITY THEORY

In this section, we discuss the identifiability of the latent states and the transition function, and provide the detailed proofs.

We assume a latent dynamical system which is viewed as high-dimensional sensory observations $\mathbf{x}_{1:T}$, where $t$ is the time point and $\mathbf{x}_t \in \mathbb{R}^D$. We assume a sequence of latent states $\mathbf{z}_{1:T}$, with $\mathbf{z}_t \in \mathbb{R}^K$, are instantaneously mapped to observations via an emission function $\mathbf{g} : \mathbb{R}^K \to \mathbb{R}^D$:

$$\mathbf{x}_t = \mathbf{g}(\mathbf{z}_t). \tag{11}$$

The latent states $\mathbf{z}_{1:T}$ evolve according to Markovian dynamics:

$$\mathbf{z}_t = \mathbf{f}(\mathbf{z}_{t-1}, \mathbf{s}_t), \tag{12}$$

where $\mathbf{f} : \mathbb{R}^{2K} \to \mathbb{R}^K$ is an auto-regressive transition function and $\mathbf{s}_t \in \mathbb{R}^K$ corresponds to process noise.

Our aim is to jointly identify the latent states $\mathbf{z}_{1:T}$, the dynamics function $\mathbf{f}$, and the process noise $\mathbf{s}_{1:T}$. We remind that previous works (Klindt et al., 2020; Yao et al., 2021; 2022; Song et al., 2023) have concentrated on identifying the latent states $\mathbf{z}_{1:T}$, possibly with linear transition approximations, but not a general transition function $\mathbf{f}$. Yet, without a general $\mathbf{f}$, the methods can estimate the underlying states only when corresponding observations are provided or provide simplistic approximations in their absence. Hence, they cannot predict complex future behavior reliably.

Notice that learning a provably identifiable transition function $\mathbf{f} : \mathbb{R}^{2K} \to \mathbb{R}^K$ is not straightforward, since the transition function is not injective. A naive solution can be to simply use a plug-in method (Yao et al., 2021; 2022) for identifying the latents and then fitting a transition function $\mathbf{f}$ on the estimated latents, however, we show empirically in our experiments that it leads to poor prediction accuracy for the future behavior.

## B.1    NONLINEAR ICA

The nonlinear ICA assumes that the data is generated from independent latent variables $\mathbf{z}$ with a nonlinear emission function $\mathbf{g}$, following Eq. (11). It is well-known to be non-identifiable for i.i.d. data

(Hyvärinen & Pajunen, 1999; Locatello et al., 2019). Recent seminal works (Hyvärinen & Morioka, 2016; 2017; Hyvärinen et al., 2019) showed that *autocorrelation* and *nonstationarity* existent in non-i.i.d. data can be exploited to identify latent variables in an unsupervised way. Compared to the vanilla ICA that considers independence only along latent dimensions, the idea of these works is to introduce additional independence constraints reflecting the existent structure in the data. These additional constraints are formulated mathematically as *identifiability assumptions*, which restrict the space of the emission function $\mathbf{g}$ and the space of the latent prior $p_{\mathbf{z}}$ (Hyvärinen et al., 2023; Xi & Bloem-Reddy, 2023). The key insight is that, after sufficiently constraining the latent prior $p_{\mathbf{z}}$ using such assumptions, identifying the latent variables $\mathbf{z}_t$ and identifying the injective emission function $\mathbf{g}$ become equivalent tasks (Xi & Bloem-Reddy, 2023).

### B.2 AUGMENTED DYNAMICS FOR IDENTIFIABLE SYSTEMS

To identify the transition function $\mathbf{f}$ such that $\mathbf{z}_t = \mathbf{f}(\mathbf{z}_{t-1}, \mathbf{s}_t)$, we will use the same insight: After sufficiently constraining the noise prior $p_{\mathbf{s}}$; given an identifiable latent pair $(\mathbf{z}_{t-1}, \mathbf{z}_t)$, identifying the noise variables $\mathbf{s}_t$ and identifying the bijective dynamics function $\mathbf{f}$ should be equivalent. Hence, in addition to the identifiability assumptions restricting the space of the emission function $\mathbf{g}$ and the space of the latent prior $p_{\mathbf{z}}$, we will further restrict the space of the dynamics function $\mathbf{f}$, and the space of the noise prior $p_{\mathbf{s}}$.

First, let us note that the identifiability of the process noise $\mathbf{s}_t$ is not trivial since the dynamics function $\mathbf{f} : \mathbb{R}^{2K} \to \mathbb{R}^K$ is not an injective function and hence it does not have an inverse. Following the independent innovation analysis (IIA) framework Morioka et al. (2021), we trivially augment the image space of the transition function and denote the bijective augmented function by $\mathbf{f}_{\mathrm{aug}} : \mathbb{R}^{2K} \to \mathbb{R}^{2K}$:

$$\begin{bmatrix} \mathbf{z}_t \\ \mathbf{z}_{t-1} \end{bmatrix} = \mathbf{f}_{\mathrm{aug}} \left( \begin{bmatrix} \mathbf{s}_t \\ \mathbf{z}_{t-1} \end{bmatrix} \right) = \begin{bmatrix} \mathbf{f}(\mathbf{z}_{t-1}, \mathbf{s}_t) \\ \mathbf{z}_{t-1} \end{bmatrix} \tag{13}$$

Contrary to Morioka et al. (2021), which use an augmented autoregressive model on *observations* $(\mathbf{x}_{t-1}, \mathbf{x}_t)$, our formulation captures the functional dependence between a *latent pair* $(\mathbf{z}_{t-1}, \mathbf{z}_t)$ and the process noise $\mathbf{s}_t$.

Next, we make the standard assumption in the temporal identifiability literature (Klindt et al., 2020; Yao et al., 2021; 2022; Song et al., 2023) that each dimension of the transition function $\{f_k\}_{k=1}^K$ is influenced by a single process noise variable $s_{kt}$. The output is a single latent variable $z_{kt}$:

$$z_{kt} = f_k(\mathbf{z}_{t-1}, s_{kt}), \quad \text{for } k \in 1, \ldots, K \text{ and } t \in 1, \ldots, T. \tag{14}$$

Notice that this does not impose a limitation on the generative model, it just creates a segmentation between noise variables and latent variables. For example, if this assumption is violated and there exists a noise variable $s_{kt}$ that affects both $z_{it}$ and $z_{jt}$ with $i \neq j$, then the noise variable $s_{kt}$ can instead be modeled as a latent variable $z_{kt}$.

We re-state the full generative model for completeness:

$$\mathbf{z}_0 \sim p_{\mathbf{z}_0}(\mathbf{z}_0), \qquad\qquad\qquad\qquad\qquad \texttt{\# initial state} \tag{15}$$

$$\mathbf{s}_t \sim p_{\mathbf{s}|\mathbf{u}}(\mathbf{s}_t|\mathbf{u}) = \prod_k p_{s_k|\mathbf{u}}(s_{kt}|\mathbf{u}), \quad \forall t \in 1, \ldots, T, \quad \texttt{\# process noise} \tag{16}$$

$$\begin{bmatrix} \mathbf{z}_t \\ \mathbf{z}_{t-1} \end{bmatrix} = \mathbf{f}_{\mathrm{aug}} \left( \begin{bmatrix} \mathbf{s}_t \\ \mathbf{z}_{t-1} \end{bmatrix} \right) = \begin{bmatrix} \mathbf{f}(\mathbf{z}_{t-1}, \mathbf{s}_t) \\ \mathbf{z}_{t-1} \end{bmatrix}, \quad \forall t \in 1, \ldots, T, \quad \texttt{\# state transition} \tag{17}$$

$$\begin{bmatrix} \mathbf{x}_t \\ \mathbf{x}_{t-1} \end{bmatrix} = \mathbf{g}_{\mathrm{aug}} \left( \begin{bmatrix} \mathbf{z}_t \\ \mathbf{z}_{t-1} \end{bmatrix} \right) = \begin{bmatrix} \mathbf{g}(\mathbf{z}_t) \\ \mathbf{g}(\mathbf{z}_{t-1}) \end{bmatrix}, \qquad \forall t \in 2, \ldots, T. \quad \texttt{\# observation mapping}$$
$$\tag{18}$$

where $\mathbf{u}$ is an auxiliary variable, which modulates the noise distribution $p_{\mathbf{s}|\mathbf{u}}$.

## C DISCUSSION OF ASSUMPTIONS

In this section, we compare our assumptions with the related work while discussing their theoretical and practical implications. To make a better connection with a real-world dynamical sytem example,

we discuss an autonomously controlled drone. Here, the observations $\mathbf{x}_{1:T}$ would be a video stream of the flying drone instead of the system state $\mathbf{z}_{1:T}$ containing absolute position, velocity, and acceleration in 3D. The process noise $\mathbf{s}_{1:T}$ might represent additional variables influencing the state evolution, e.g., the strength and direction of the wind at any time or drone motor torque set by the controller. In the next paragraph, we specifically consider the latent state $\mathbf{z}_t$ as the 3D position (location and orientation) of the drone at time $t$, and the external noise $\mathbf{s}_t$ as wind affecting the drone.

Here, we refer to the observational equivalence of the models as Assumption *(A0)*, i.e., the ground-truth and the learned densities are observationally equivalent: $p_{\mathbf{f}_{\mathrm{aug}}, \mathbf{g}_{\mathrm{aug}}, p_{\mathbf{s}|\mathbf{u}}}(\{\mathbf{x}_t\}_{t=1}^T) = p_{\hat{\mathbf{f}}_{\mathrm{aug}}, \hat{\mathbf{g}}_{\mathrm{aug}}, \hat{p}_{\mathbf{s}|\mathbf{u}}}(\{\mathbf{x}_t\}_{t=1}^T), \forall \mathbf{x}_t \in \mathcal{X}$ for $t \in 1, \ldots, T$.

Assumptions *(A0,A1,A2)* are standard assumptions in the nonlinear ICA literature (Klindt et al., 2020; Morioka et al., 2021; Yao et al., 2021; 2022). Assumption *(A0)* ensures that the model $\hat{\mathcal{M}}$ is sufficiently flexible that it learns the correct distribution in the limit of infinite data, e.g., the learned model components are universal approximators (neural networks), or for a variational inference algorithm the family of the variational posterior contains the true posterior. For the drone example, most likely the drone dynamics is nonlinear, and hence modeling the transition function $\mathbf{f}$ by a linear function would violate this assumption. In Assumption *(A1)*, the injectivity of the augmented emission function implies that we have one-to-one mapping between the latent states and the observation manifold. In the drone example, this implies that the 3D position should map to a unique drone position in the pixel space. This could be violated in natural videos as they exhibit complex dependencies such as occlusions. The bijectivity of the augmented latent transition function captures the functional dependence between a *latent pair* $(\mathbf{z}_{t-1}, \mathbf{z}_t)$ and the process noise $\mathbf{s}_t$, in contrast to Morioka et al. (2021), which use an augmented transition model on *observations* $(\mathbf{x}_{t-1}, \mathbf{x}_t)$. For the drone example, for a given 3D position $\mathbf{z}_{t-1}$, the transition function $\mathbf{f}$ must map two distinct wind intensities $\mathbf{s}_t$ to distinct 3D positions $\mathbf{z}_t$. This assumption would not hold, e.g., if the physical system that applies the torque set by the controller has some intrinsic noise and the transition function $\mathbf{f}$ is stochastic. Assumptions *(A2, A3, A4)* generalize the nonstationary noise and the sufficient variability assumptions in Yao et al. (2021) to a general auxiliary variable $\mathbf{u}$. For the drone example, they imply that the underlying system can be decomposed into (conditionally) independent 1D variables, e.g., the 3D position of the drone to 1D variables $x, y, z$, which are affected independently by the $x, y, z$ values of the wind intensity (noise). Assumptions *(A4, A5)* imply that the latent states and the process noise show enough non-i.i.d. structure in the form of autocorrelation or non-stationarity, i.e., they change sufficiently differently with respect to $2K$ values of $\mathbf{u}$: $\mathbf{u}_1, \ldots, \mathbf{u}_{2K}$ (Hyvärinen et al., 2019; 2023). For *(A5)* to hold, we need to observe the drone in sufficiently many environments with sufficiently different wind (noise) conditions. For *(A4)* to hold, the changes in the wind distribution should sufficiently change the distribution of 3D positions (latent states). These assumptions are violated, e.g., if the wind is constant across the observed environments.

**Remark 1** Yao et al. (2021) used nonstationarity of the process noise for identifiability of the conditionally independent latent states (**Theorem 1**). If the variable $\mathbf{u}$ is an observed categorical variable (e.g., domain indicator), the assumptions *(A4, A5)* can be written in an alternative form without partial derivatives with respect to $u_l$ (Hyvärinen et al., 2019) (see App. C.3 for the alternative version).

**Remark 2** In the presence of control inputs, we can write the augmented dynamics function given the control input as $\mathbf{f}_{\mathrm{aug}}(\cdot|a) : \mathbb{R}^{2K} \to \mathbb{R}^{2K}$. The identifiability proof requires that the augmented dynamics function to be bijective. Then, the identifiability proof continues to hold as long as the augmented dynamics function conditioned for a given control input $a$, $\mathbf{f}_{\mathrm{aug}}(\cdot|a) : \mathbb{R}^{2K} \to \mathbb{R}^{2K}$, is also bijective for any given $a$.

## C.1 PROOF OF THEOREM 1: IDENTIFIABILITY OF THE LATENT STATES $\mathbf{z}_t$

This result is already shown in (Yao et al., 2021, Appendix A.3.2). Here, we follow Klindt et al. (2020); Yao et al. (2021; 2022) and repeat their results in our notation as we also make use of this result in App. C.2.

The injective functions $\mathbf{g}, \hat{\mathbf{g}} : \mathbb{R}^K \to \mathbb{R}^D$ are bijective between the latent space $\mathbb{R}^K$ and the observation space $\mathcal{X} \subset \mathbb{R}^D$. We denote the inverse functions from the restricted observation space to

the latent space by $\mathbf{g}^{-1}, \hat{\mathbf{g}}^{-1}$. This is also implicitly assumed in (Klindt et al., 2020; Yao et al., 2021; 2022; Song et al., 2023).

Using $\mathbf{g}$ and $\hat{\mathbf{g}}$, we can relate the ground-truth and estimated latents $\mathbf{z}_t$ and $\hat{\mathbf{z}}_t$ to each other:

$$\mathbf{x}_t = \hat{\mathbf{g}}(\hat{\mathbf{z}}_t) = \left( (\mathbf{g} \circ \underbrace{\mathbf{g}^{-1}) \circ \hat{\mathbf{g}}}_{\mathbf{h}} \right)(\hat{\mathbf{z}}_t) \implies \hat{\mathbf{g}} = \mathbf{g} \circ \mathbf{h} \implies \mathbf{z}_t = \mathbf{h}(\hat{\mathbf{z}}_t), \tag{19}$$

where the function $\mathbf{h} : \hat{\mathbf{z}}_t \mapsto \mathbf{z}_t$ maps the learned latents to the ground-truth latents. To show it is bijective, we need to show it is both injective and surjective. Following Klindt et al. (2020), it is injective since it is a composition of injective functions. Assume it is not surjective, then there exists a neighborhood $\mathbf{U}_\mathbf{z}$ for which $\mathbf{g}(\mathbf{U}_\mathbf{z}) \notin \hat{\mathbf{g}}(\mathbb{R}^K)$. This implies that the neighborhood of images generated by $\mathbf{g}(\mathbf{U}_\mathbf{z})$ has zero density under the learned observation density $p_{\hat{\mathbf{g}}_{\mathrm{aug}}, \hat{\mathbf{f}}_{\mathrm{aug}}, \hat{p}_{\mathbf{s}|\mathbf{u}}}(\mathbf{g}(\mathbf{U}_\mathbf{z})) = 0$, while having non-zero density under the ground-truth observation density $p_{\mathbf{g}, \mathbf{f}_{\mathrm{aug}}}(\mathbf{x})$: $p_{\mathbf{g}_{\mathrm{aug}}, \mathbf{f}_{\mathrm{aug}}, p_{\mathbf{s}|\mathbf{u}}}(\mathbf{g}(\mathbf{U}_\mathbf{z})) > 0$. This contradicts the assumption that the observation densities match everywhere. Then, $\mathbf{h}$ is surjective.

Using $\mathbf{z}_t = \mathbf{h}(\hat{\mathbf{z}}_t)$, we can analyze how the conditional densities of the ground-truth and estimated latents $\mathbf{z}_t$ and $\hat{\mathbf{z}}_t$ are related. For this, we perform change of variables on the conditional latent density $\log p(\hat{\mathbf{z}}_t|\hat{\mathbf{z}}_{t-1}, \mathbf{u})$ as follows:

$$\log p(\hat{\mathbf{z}}_t|\hat{\mathbf{z}}_{t-1}, \mathbf{u}) = \log p(\mathbf{z}_t|\mathbf{z}_{t-1}, \mathbf{u}) + \log |\mathbf{H}_t|, \tag{20}$$

$$\sum_{k=1}^K \underbrace{\log p(\hat{z}_{kt}|\hat{\mathbf{z}}_{t-1}, \mathbf{u})}_{\hat{\eta}_k(\hat{z}_{kt}, \mathbf{u})} = \sum_{k=1}^K \underbrace{\log p(z_{kt}|\mathbf{z}_{t-1}, \mathbf{u})}_{\eta_k(z_{kt}, \mathbf{u})} + \log |\mathbf{H}_t| \tag{21}$$

$$\sum_{k=1}^K \hat{\eta}_k(\hat{z}_{kt}, \mathbf{u}) = \sum_{k=1}^K \eta_k(z_{kt}, \mathbf{u}) + \log |\mathbf{H}_t| \tag{22}$$

where $\mathbf{H}_t = \mathbf{J}_\mathbf{h}(\hat{\mathbf{z}}_t)$ is the Jacobian matrix of $\mathbf{h}$ evaluated at $\hat{\mathbf{z}}_t$. When we take derivatives of both sides with respect to $\hat{z}_{it}$ in Eq. (22), the left-hand side reduces to the single term since the conditional density of the estimated latents factorize:

$$\frac{\partial \hat{\eta}_i(\hat{z}_{it}, \mathbf{u})}{\partial \hat{z}_{it}} = \sum_{k=1}^K \frac{\partial \eta_k(z_{kt}, \mathbf{u})}{\partial z_{kt}} \frac{\partial z_{kt}}{\partial \hat{z}_{it}} + \frac{\partial \log |\mathbf{H}_t|}{\partial \hat{z}_{it}}. \tag{23}$$

Next, we take derivatives with respect to $\hat{z}_{jt}$ in Eq. (23), and the left-hand side becomes $0$ since a term $\hat{\eta}_i(\hat{z}_{it}, \mathbf{u})$ does not depend on $\hat{z}_{jt}$.:

$$0 = \sum_{k=1}^K \left( \frac{\partial^2 \eta_k(z_{kt}, \mathbf{u})}{\partial z_{kt}^2} \frac{\partial z_{kt}}{\partial \hat{z}_{it}} \frac{\partial z_{kt}}{\partial \hat{z}_{jt}} + \frac{\partial \eta_k(z_{kt}, \mathbf{u})}{\partial z_{kt}} \frac{\partial z_{kt}^2}{\partial \hat{z}_{it} \partial \hat{z}_{jt}} \right) + \frac{\partial \log |\mathbf{H}_t|}{\partial \hat{z}_{it} \partial \hat{z}_{jt}}. \tag{24}$$

Lastly, take derivatives with respect to $u_l$ in Eq. (24):

$$0 = \sum_{k=1}^K \left( \frac{\partial^3 \eta_k(z_{kt}, \mathbf{u})}{\partial z_{kt}^2 \partial u_l} \frac{\partial z_{kt}}{\partial \hat{z}_{it}} \frac{\partial z_{kt}}{\partial \hat{z}_{jt}} + \frac{\partial^2 \eta_k(z_{kt}, \mathbf{u})}{\partial z_{kt} \partial u_l} \frac{\partial z_{kt}^2}{\partial \hat{z}_{it} \partial \hat{z}_{jt}} \right), \tag{25}$$

$$= \sum_{k=1}^K \left( \frac{\partial^3 \eta_k(z_{kt}, \mathbf{u})}{\partial z_{kt}^2 \partial u_l} [\mathbf{H}_t]_{ki} [\mathbf{H}_t]_{kj} + \frac{\partial^2 \eta_k(z_{kt}, \mathbf{u})}{\partial z_{kt} \partial u_l} \frac{\partial z_{kt}^2}{\partial \hat{z}_{it} \partial \hat{z}_{jt}} \right), \tag{26}$$

since the Jacobian $\mathbf{H}_t$ does not depend on $\mathbf{u}$. Using the sufficient variability assumption *(A3)* for the latent states $\mathbf{z}_t$, we can plug in $2K$ values of $\mathbf{u}_1, \ldots, \mathbf{u}_{2K}$ for which the partial derivatives of the log conditional density $\eta_k(z_{kt}, \mathbf{u})$ form linearly independent vectors $\mathbf{v}(\mathbf{z}_t, \mathbf{u})$. We see that the coefficients of these linearly independent vectors have to be zero: $[\mathbf{H}_t]_{ki}[\mathbf{H}_t]_{kj} = 0$. This implies that the Jacobian matrix $\mathbf{H}_t$ of the transformation $\mathbf{z}_t = \mathbf{h}(\hat{\mathbf{z}}_t)$ has at most $1$ nonzero element in its rows. Therefore, the learned latents $\hat{\mathbf{z}}_t$ are equivalent to the ground-truth latents $\mathbf{z}_t$ up to permutations and invertible, element-wise nonlinear transformations.

## C.2 PROOF OF THEOREM 2: IDENTIFIABILITY OF THE LATENT TRANSITION $\mathbf{f}$

In this section, we prove our main theoretical contribution, **Theorem 2**. We first define the function that relates the ground-truth and estimated latent pairs $(\mathbf{s}_t, \mathbf{z}_{t-1})$ and $(\hat{\mathbf{s}}_t, \hat{\mathbf{z}}_t)$ to each other. Similar

to the proof of **Theorem 1**, we can do this by using the relationship between $(\mathbf{f}_{\text{aug}}, \mathbf{g}_{\text{aug}})$ and $(\hat{\mathbf{f}}_{\text{aug}}, \hat{\mathbf{g}}_{\text{aug}})$:

$$\begin{bmatrix} \mathbf{x}_t \\ \mathbf{x}_{t-1} \end{bmatrix} = (\hat{\mathbf{g}}_{\text{aug}} \circ \hat{\mathbf{f}}_{\text{aug}}) \left( \begin{bmatrix} \hat{\mathbf{s}}_t \\ \hat{\mathbf{z}}_{t-1} \end{bmatrix} \right) = \begin{bmatrix} (\hat{\mathbf{g}} \circ \hat{\mathbf{f}})(\hat{\mathbf{z}}_{t-1}, \hat{\mathbf{s}}_t) \\ \hat{\mathbf{g}}(\hat{\mathbf{z}}_{t-1}) \end{bmatrix} \tag{27}$$

$$= (\mathbf{g}_{\text{aug}} \circ \mathbf{f}_{\text{aug}}) \circ \underbrace{(\mathbf{g}_{\text{aug}} \circ \mathbf{f}_{\text{aug}})^{-1} \circ (\hat{\mathbf{g}}_{\text{aug}} \circ \hat{\mathbf{f}}_{\text{aug}})}_{\mathbf{k}_{\text{aug}}} \left( \begin{bmatrix} \hat{\mathbf{s}}_t \\ \hat{\mathbf{z}}_{t-1} \end{bmatrix} \right) \tag{28}$$

Then, the function $\mathbf{k}_{\text{aug}} : \mathbb{R}^{2K} \to \mathbb{R}^{2K}$ maps the learned pair $(\hat{\mathbf{s}}_t, \hat{\mathbf{z}}_{t-1})$ to the ground-truth pair $(\mathbf{s}_t, \mathbf{z}_{t-1})$:

$$\begin{bmatrix} \mathbf{s}_t \\ \mathbf{z}_{t-1} \end{bmatrix} = \mathbf{k}_{\text{aug}} \left( \begin{bmatrix} \hat{\mathbf{s}}_t \\ \hat{\mathbf{z}}_{t-1} \end{bmatrix} \right) = \begin{bmatrix} \mathbf{k}_1(\hat{\mathbf{s}}_t, \hat{\mathbf{z}}_{t-1}) \\ \mathbf{k}_2(\hat{\mathbf{s}}_t, \hat{\mathbf{z}}_{t-1}) \end{bmatrix}. \tag{29}$$

Similar to the function $\mathbf{h}$ being bijective in App. C.1, it follows that $\mathbf{k}_{\text{aug}}$ is also bijective.

Now, we want to show that the augmented function $\mathbf{k}_{\text{aug}}$ decomposes into invertible block-wise functions $\mathbf{k}_1, \mathbf{k}_2 : \mathbb{R}^K \to \mathbb{R}^K$ such that (i) $\mathbf{k}_1$ only depends on $\hat{\mathbf{s}}_t$, i.e., $\mathbf{s}_t = \mathbf{k}_1(\hat{\mathbf{s}}_t)$ and (ii) $\mathbf{k}_2$ only depends on $\hat{\mathbf{z}}_{t-1}$, i.e., $\mathbf{z}_{t-1} = \mathbf{k}_2(\hat{\mathbf{z}}_{t-1})$. It is easy to show (ii) the second case, since $\mathbf{k}_2 = \text{id}_\mathbf{z} \circ \mathbf{g}^{-1} \circ \hat{\mathbf{g}} \circ \text{id}_{\hat{\mathbf{z}}} = \mathbf{h}$ and we have already shown in App. C.1 that the function $\mathbf{h} : \hat{\mathbf{z}}_t \mapsto \mathbf{z}_t$ is equal to $\mathbf{h} = \mathbf{g}^{-1} \circ \hat{\mathbf{g}}$ and bijective. Hence, we can write $\mathbf{h}(\hat{\mathbf{z}}_{t-1})$ in the place of $\mathbf{k}_2(\hat{\mathbf{s}}_t, \hat{\mathbf{z}}_{t-1})$:

$$\begin{bmatrix} \mathbf{s}_t \\ \mathbf{z}_{t-1} \end{bmatrix} = \mathbf{k}_{\text{aug}} \left( \begin{bmatrix} \hat{\mathbf{s}}_t \\ \hat{\mathbf{z}}_{t-1} \end{bmatrix} \right) = \begin{bmatrix} \mathbf{k}_1(\hat{\mathbf{s}}_t, \hat{\mathbf{z}}_{t-1}) \\ \mathbf{h}(\hat{\mathbf{z}}_{t-1}) \end{bmatrix}. \tag{30}$$

To show (i), we start by performing change of variables on the log density of the pair $(\hat{\mathbf{s}}_t, \hat{\mathbf{z}}_{t-1})$ for the transformation $\mathbf{k}_{\text{aug}} : (\hat{\mathbf{s}}_t, \hat{\mathbf{z}}_{t-1}) \mapsto (\mathbf{s}_t, \mathbf{z}_{t-1})$:

$$\log p(\hat{\mathbf{s}}_t, \hat{\mathbf{z}}_{t-1}|\mathbf{u}) = \log p(\mathbf{k}_{\text{aug}}(\hat{\mathbf{s}}_t, \hat{\mathbf{z}}_{t-1})|\mathbf{u}) + \log |\mathbf{J}_{\mathbf{k}_{\text{aug}}}(\hat{\mathbf{s}}_t, \hat{\mathbf{z}}_{t-1})|, \tag{31}$$

$$= \log p([\mathbf{k}_1(\hat{\mathbf{s}}_t, \hat{\mathbf{z}}_{t-1}), \mathbf{h}(\hat{\mathbf{z}}_{t-1})]|\mathbf{u}) + \log |\mathbf{J}_{\mathbf{k}_{\text{aug}}}(\hat{\mathbf{s}}_t, \hat{\mathbf{z}}_{t-1})|, \tag{32}$$

where $\mathbf{J}_{\mathbf{k}_{\text{aug}}}(\hat{\mathbf{s}}_t, \hat{\mathbf{z}}_{t-1})$ is the Jacobian matrix for the augmented function $\mathbf{k}_{\text{aug}}$ evaluated at $(\hat{\mathbf{s}}_t, \hat{\mathbf{z}}_{t-1})$. As the process noise is temporally independent given $\mathbf{u}$: $\hat{\mathbf{s}}_t \perp\!\!\!\perp \hat{\mathbf{z}}_{t-1}|\mathbf{u}$ and $\mathbf{s}_t \perp\!\!\!\perp \mathbf{z}_{t-1}|\mathbf{u}$, we factorize the densities in Eq. (32):

$$\log p(\hat{\mathbf{s}}_t|\mathbf{u}) + \log p(\hat{\mathbf{z}}_{t-1}|\mathbf{u}) = \log p(\mathbf{k}_1(\hat{\mathbf{s}}_t, \hat{\mathbf{z}}_{t-1})|\mathbf{u}) + \log p(\mathbf{h}(\hat{\mathbf{z}}_{t-1})|\mathbf{u}) + \log |\mathbf{J}_{\mathbf{k}_{\text{aug}}}(\hat{\mathbf{s}}_t, \hat{\mathbf{z}}_{t-1})|. \tag{33}$$

The Jacobian $\mathbf{J}_{\mathbf{k}_{\text{aug}}}$ is upper block-diagonal since $\mathbf{z}_{t-1}$ does not depend on $\hat{\mathbf{s}}_t$: $\mathbf{J}_{\mathbf{k}_{\text{aug}}} = \begin{bmatrix} \frac{\partial \mathbf{s}_t}{\partial \hat{\mathbf{s}}_t} & * \\ \mathbf{0} & \mathbf{H}_t \end{bmatrix}$.

Hence, its log determinant factorizes as $\log |\mathbf{J}_{\mathbf{k}_{\text{aug}}}(\hat{\mathbf{s}}_t, \hat{\mathbf{z}}_{t-1})| = \log |\mathbf{H}_t| + \log |\frac{\partial \mathbf{s}_t}{\partial \hat{\mathbf{s}}_t}|$. We can add this factorization into Eq. (33):

$$\log p(\hat{\mathbf{s}}_t|\mathbf{u}) + \log p(\hat{\mathbf{z}}_{t-1}|\mathbf{u}) = \log p(\mathbf{k}_1(\hat{\mathbf{s}}_t, \hat{\mathbf{z}}_{t-1})|\mathbf{u}) + \log p(\mathbf{h}(\hat{\mathbf{z}}_{t-1})|\mathbf{u}) + \log |\mathbf{H}_t| + \log |\frac{\partial \mathbf{s}_t}{\partial \hat{\mathbf{s}}_t}|. \tag{34}$$

In addition, the noise is conditionally independent over its dimensions given $\mathbf{u}$. Therefore, we can further factorize the noise densities $p(\hat{\mathbf{s}}_t|\mathbf{u}) = \prod_k p(\hat{s}_{kt}|\mathbf{u})$ and $p(\mathbf{k}_1(\hat{\mathbf{s}}_t, \hat{\mathbf{z}}_{t-1})|\mathbf{u}) = p(\mathbf{s}_t|\mathbf{u}) = \prod_k p(s_{kt}|\mathbf{u})$ with $\mathbf{s}_t = \mathbf{k}_1(\hat{\mathbf{s}}_t, \hat{\mathbf{z}}_{t-1})$. We incorporate these factorizations into Eq. (34) as follows:

$$\sum_k \underbrace{\log p(\hat{s}_{kt}|\mathbf{u})}_{\hat{q}_k(\hat{s}_{kt}, \mathbf{u})} + \log p(\hat{\mathbf{z}}_{t-1}|\mathbf{u}) = \sum_k \underbrace{\log p(s_{kt}|\mathbf{u})}_{q_k(s_{kt}, \mathbf{u})} + \log p(\mathbf{h}(\hat{\mathbf{z}}_{t-1})|\mathbf{u}) + \log |\mathbf{H}_t| + \log |\frac{\partial \mathbf{s}_t}{\partial \hat{\mathbf{s}}_t}|, \tag{35}$$

When we take the derivative of both sides of Eq. (35) with respect to $\hat{s}_{it}$, the term $\log p(\hat{\mathbf{z}}_{t-1}|\mathbf{u})$ at the left-hand side, and the terms $\log p(\mathbf{h}(\hat{\mathbf{z}}_{t-1})|\mathbf{u})$ and $\log |\mathbf{H}_t|$ vanish as they do not depend on $\hat{s}_{it}$:

$$\frac{\partial \hat{q}_i(\hat{s}_{it}, \mathbf{u})}{\partial \hat{s}_{it}} = \sum_k \frac{\partial q_k(s_{kt}, \mathbf{u})}{\partial s_{kt}} \frac{\partial s_{kt}}{\partial \hat{s}_{it}} + \frac{\partial \log |\frac{\partial \mathbf{s}_t}{\partial \hat{\mathbf{s}}_t}|}{\partial \hat{s}_{it}}. \tag{36}$$

Next, we take the derivative with respect to $u_l$ with $l$ being an arbitrary dimension, and the term $\frac{\partial \log |\frac{\partial \mathbf{s}_t}{\partial \hat{\mathbf{s}}_t}|}{\partial \hat{s}_{it}}$ vanishes since $|\frac{\partial \mathbf{s}_t}{\partial \hat{\mathbf{s}}_t}|$ does not depend on $\mathbf{u}$:

$$\frac{\partial^2 \hat{q}_i(\hat{s}_{it}, \mathbf{u})}{\partial \hat{s}_{it} \partial u_l} = \sum_k \frac{\partial^2 q_k(s_{kt}, \mathbf{u})}{\partial s_{kt} \partial u_l} \frac{\partial s_{kt}}{\partial \hat{s}_{it}}. \tag{37}$$

Lastly, take the derivative of both sides with respect to $\hat{z}_{j,t-1}$:

$$0 = \sum_k \left( \frac{\partial^3 q_k(s_{kt}, \mathbf{u})}{\partial s_{kt}^2 \partial u_l} \frac{\partial s_{kt}}{\partial \hat{s}_{it}} \frac{\partial s_{kt}}{\partial \hat{z}_{j,t-1}} + \frac{\partial^2 q_k(s_{kt}, \mathbf{u})}{\partial s_{kt} \partial u_l} \frac{\partial^2 s_{kt}}{\partial \hat{s}_{it} \partial \hat{z}_{j,t-1}} \right). \tag{38}$$

Inspecting the Eq. (38), to ensure the sufficient variability assumption *(A4)* for the process noise $\mathbf{s}_t$, the term $\frac{\partial s_{kt}}{\partial \hat{s}_{it}} \frac{\partial s_{kt}}{\partial \hat{z}_{j,t-1}} = 0$. Following a similar reasoning with Morioka et al. (2021), this implies that any dimension $k$ of $\mathbf{s}_t$ does not depend on both $\hat{\mathbf{s}}_t$ and $\hat{\mathbf{z}}_{t-1}$ at the same time. Since $\mathbf{s}_t \perp\!\!\!\perp \mathbf{z}_{t-1}|\mathbf{u}$ and $\mathbf{z}_{t-1} = \mathbf{h}(\hat{\mathbf{z}}_{t-1})$, $\mathbf{s}_t$ has to depend solely on $\hat{\mathbf{s}}_t$: $\mathbf{s}_t = \mathbf{k}_1(\hat{\mathbf{z}}_{t-1}, \hat{\mathbf{s}}_t) = \mathbf{k}(\hat{\mathbf{s}}_t)$. This concludes that the augmented function $\mathbf{k}_{\text{aug}}$ decomposes into invertible block-wise functions $\mathbf{k}$ and $\mathbf{h}$: $\mathbf{k}_{\text{aug}} = [\mathbf{k}, \mathbf{h}]$:

$$\begin{bmatrix} \mathbf{s}_t \\ \mathbf{z}_{t-1} \end{bmatrix} = \mathbf{k}_{\text{aug}} \left( \begin{bmatrix} \hat{\mathbf{s}}_t \\ \hat{\mathbf{z}}_{t-1} \end{bmatrix} \right) = \begin{bmatrix} \mathbf{k}(\hat{\mathbf{s}}_t) \\ \mathbf{h}(\hat{\mathbf{z}}_{t-1}) \end{bmatrix}. \tag{39}$$

We still need to show that the function $\mathbf{k}$ that relates the ground-truth and estimated noise variables $\mathbf{s}_t$ and $\hat{\mathbf{s}}_t$, $\mathbf{s}_t = \mathbf{k}(\hat{\mathbf{s}}_t)$, is an element-wise function. For this, we get back to Eq. (37). We denote the Jacobian matrix of function $\mathbf{k}$ by $\mathbf{J}_\mathbf{k}$ and its evaluation at $\hat{\mathbf{s}}_t$ by $\mathbf{J}_\mathbf{k}(\hat{\mathbf{s}}_t) = \mathbf{K}_t$. We take the derivative of both sides in Eq. (37) with respect to $\hat{s}_{mt}$ for some index $m$:

$$0 = \sum_k \left( \frac{\partial^3 q_k(s_{kt}, \mathbf{u})}{\partial s_{kt}^2 \partial u_l} [\mathbf{K}_t]_{ki} [\mathbf{K}_t]_{km} + \frac{\partial^2 q_k(s_{kt}, \mathbf{u})}{\partial s_{kt} \partial u_l} \frac{\partial^2 s_{kt}}{\partial \hat{s}_{it} \partial \hat{s}_{mt}} \right). \tag{40}$$

Now, inspecting the Eq. (40), we see that to ensure the sufficient variability assumption *(A4)*, the product $[\mathbf{K}_t]_{ki}[\mathbf{K}_t]_{km} = 0$. This implies that each dimension $s_{kt}$ of the true process noise depends only on a single dimension of the learned process noise $\hat{\mathbf{s}}_t$. Hence, the function $\mathbf{k}$ is equal to a composition of permutation and element-wise, invertible nonlinear transformation: $\mathbf{k} = \pi \circ T$.

As the final step, we follow Eq. (28) and write the relationship between the augmented functions as:

$$\mathbf{g}_{\text{aug}} \circ \mathbf{f}_{\text{aug}} \circ \mathbf{k}_{\text{aug}} = \hat{\mathbf{g}}_{\text{aug}} \circ \hat{\mathbf{f}}_{\text{aug}}, \tag{41}$$

$$\underbrace{\hat{\mathbf{g}}_{\text{aug}}^{-1} \circ \mathbf{g}_{\text{aug}}}_{\mathbf{h}_{\text{aug}}^{-1}} \circ \mathbf{f}_{\text{aug}} \circ \mathbf{k}_{\text{aug}} = \hat{\mathbf{f}}_{\text{aug}}, \tag{42}$$

$$\mathbf{h}_{\text{aug}}^{-1} \circ \mathbf{f}_{\text{aug}} \circ \mathbf{k}_{\text{aug}} = \hat{\mathbf{f}}_{\text{aug}}, \tag{43}$$

where $\mathbf{h}_{\text{aug}}$ denotes simply the concatenation $[\mathbf{h}, \mathbf{h}]$, and hence we have $\mathbf{h}_{\text{aug}}^{-1} = [\mathbf{h}^{-1}, \mathbf{h}^{-1}]$. We have already shown that both $\mathbf{h}$ and $\mathbf{k}$ are compositions of permutations and element-wise invertible transformations. Inspecting Eq. (43), we see that the augmented transition function $\hat{\mathbf{f}}_{\text{aug}}$ and the true augmented transition function $\mathbf{f}_{\text{aug}}$ are related to each other only through functions $\mathbf{h}$ and $\mathbf{k}$ which are compositions of permutations and element-wise invertible transformations. Hence, we conclude that the augmented transition function $\hat{\mathbf{f}}_{\text{aug}}$ is equivalent to the true augmented transition function $\mathbf{f}_{\text{aug}}$ up to compositions of permutations and element-wise transformations.

## C.3 Alternative Versions of Sufficient Variability Assumption

If the variable $\mathbf{u}$ is an observed categorical variable (e.g., domain indicator), the assumptions *(A4, A5)* can be written in an alternative form without partial derivatives with respect to $u_l$, similar to Hyvärinen et al. (2019); Yao et al. (2021). For example, for the latent states $\mathbf{z}_t$, the alternative version of the *(A4)* takes the form:

- **Sufficient variability of latent states for a categorical $\mathbf{u}$ (Yao et al., 2021).** For any $\mathbf{z}_t$, there exist some $2K + 1$ values for $\mathbf{u}$: $\mathbf{u}_1, \ldots, \mathbf{u}_{2K}$, such that the $2K$ vectors $\mathbf{v}(\mathbf{z}_t, \mathbf{u}_{j+1}) - \mathbf{v}(\mathbf{z}_t, \mathbf{u}_j)$ with $j = 0, 1, \ldots, 2K$, are linearly independent where

$$\mathbf{v}(\mathbf{z}_t, \mathbf{u}) = \left( \frac{\partial \eta_1(z_{1t}, \mathbf{u})}{\partial z_{1t}}, \cdots, \frac{\partial \eta_K(z_{Kt}, \mathbf{u})}{\partial z_{Kt}}, \frac{\partial^2 \eta_1(z_{1t}, \mathbf{u})}{\partial z_{1t}^2}, \cdots, \frac{\partial^2 \eta_K(z_{Kt}, \mathbf{u})}{\partial z_{Kt}^2} \right) \in \mathbb{R}^{2K}. \tag{44}$$

A similar categorical version is provided in (Hyvärinen et al., 2019, Assumption 3), while the continuous version is provided in the same work (Hyvärinen et al., 2019, Appendix D).

# D  VARIATIONAL INFERENCE

Similar to previous works (Yao et al., 2021; 2022), we want to maximize the marginal log-likelihood $\log p(\mathbf{x}_{1:T}|\mathbf{u})$ that is obtained by marginalizing over the latent states $\mathbf{z}_{1:T}$ and process noise $\mathbf{s}_{1:T}$:

$$\log p_\theta(\mathbf{x}_{1:T}|\mathbf{u}) = \log \int_{\mathbf{z},\mathbf{s}} p_\theta(\mathbf{x}_{1:T}, \mathbf{z}_{0:T}, \mathbf{s}_{1:T}|\mathbf{u})\, \mathrm{d}\mathbf{z}_{0:T}\, \mathrm{d}\mathbf{s}_{1:T}, \tag{45}$$

where we decompose the joint distribution as follows:

$$p_\theta(\mathbf{x}_{1:T}, \mathbf{z}_{0:T}, \mathbf{s}_{1:T}|\mathbf{u}) = p_\theta(\mathbf{z}_0) \prod_{t=1}^{T} p_\theta(\mathbf{s}_t|\mathbf{u}) \underbrace{p_\theta(\mathbf{z}_t|\mathbf{z}_{t-1}, \mathbf{s}_t)}_{\delta(\mathbf{z}_t - \mathbf{f}(\mathbf{s}_t, \mathbf{z}_{t-1}))} p_\theta(\mathbf{x}_t|\mathbf{z}_t). \tag{46}$$

Note that the state transitions $p_\theta(\mathbf{z}_t|\mathbf{z}_{t-1}, \mathbf{s}_t)$ are assumed to be deterministic. The above integral is intractable due to non-linear dynamics $\mathbf{f}$ and observation $\mathbf{g}$ functions. As typically done with the deep latent variable models, we approximate the log marginal likelihood by a variational lower bound, i.e., we introduce an amortized approximate posterior distribution $q_\phi(\mathbf{z}_{0:T}, \mathbf{s}_{1:T}|\mathbf{x}_{1:T}, \mathbf{u})$ that decomposes as follows:

$$q_\phi(\mathbf{z}_{0:T}, \mathbf{s}_{1:T}|\mathbf{x}_{1:T}, \mathbf{u}) = q_\phi(\mathbf{z}_0|\mathbf{x}_{1:T}, \mathbf{u}) \prod_{t=1}^{T} \underbrace{q_\phi(\mathbf{s}_t|\mathbf{z}_{0:t-1}, \mathbf{s}_{1:t-1}, \mathbf{x}_{1:T}, \mathbf{u})}_{q_\phi(\mathbf{s}_t|\mathbf{z}_{t-1}, \mathbf{x}_{1:t})} \underbrace{q_\phi(\mathbf{z}_t|\mathbf{z}_{0:t-1}, \mathbf{s}_{1:t}, \mathbf{u})}_{q_\phi(\mathbf{z}_t|\mathbf{z}_{t-1}, \mathbf{s}_t)}$$
$$\tag{47}$$

We simplify the variational posterior $q(\mathbf{s}_t|\cdot)$ as $q_\phi(\mathbf{s}_t|\mathbf{z}_{0:t-1}, \mathbf{s}_{1:t-1}, \mathbf{x}_{1:T}, \mathbf{u}) = q_\phi(\mathbf{s}_t|\mathbf{z}_{t-1}, \mathbf{x}_{1:t})$, corresponding to a filtering distribution. As in the generative model, we choose $q_\phi(\mathbf{z}_t|\mathbf{z}_{t-1}, \mathbf{s}_t) = p_\theta(\mathbf{z}_t|\mathbf{z}_{t-1}, \mathbf{s}_t) = \delta(\mathbf{z}_t - \mathbf{f}(\mathbf{s}_t, \mathbf{z}_{t-1}))$:

$$q_\phi(\mathbf{z}_{0:T}, \mathbf{s}_{1:T}|\mathbf{x}_{1:T}, \mathbf{u}) = q_\phi(\mathbf{z}_0|\mathbf{x}_{1:T}) \prod_{t=1}^{T} q_\phi(\mathbf{s}_t|\mathbf{z}_{t-1}, \mathbf{x}_{1:t}) p_\theta(\mathbf{z}_t|\mathbf{z}_{t-1}, \mathbf{s}_t), \tag{48}$$

where the functional forms of the densities $q_\phi(\mathbf{z}_0|\cdot)$ and $q_\phi(\mathbf{s}_t|\cdot)$ are chosen as diagonal Gaussian distributions whose parameters are computed by recurrent neural networks. The variational lower bound takes the following form:

$$\mathcal{L}(\theta, \phi) = \mathbb{E}_{q_\phi(\mathbf{z}_{0:T}, \mathbf{s}_{1:T})} \left[ \log p_\theta(\mathbf{x}_{1:T}|\mathbf{z}_{1:T}) + \log \frac{p_\theta(\mathbf{z}_{0:T}, \mathbf{s}_{1:T}|\mathbf{u})}{q_\phi(\mathbf{z}_{0:T}, \mathbf{s}_{1:T}|\mathbf{x}_{1:T}, \mathbf{u})} \right] \tag{49}$$

$$= \underbrace{\sum_{t=1}^{T} \mathbb{E}_{q_\phi(\mathbf{z}_t)}[\log p_\theta(\mathbf{x}_t|\mathbf{z}_t)]}_{\text{Reconstruction term, } \mathcal{L}_R} + \underbrace{\mathbb{E}_{q_\phi(\mathbf{z}_{0:T}, \mathbf{s}_{1:T})} \left[ \log \frac{p_\theta(\mathbf{z}_{0:T}, \mathbf{s}_{1:T}|\mathbf{u})}{q_\phi(\mathbf{z}_{0:T}, \mathbf{s}_{1:T}|\mathbf{x}_{1:T}, \mathbf{u})} \right]}_{\text{KL term, } \mathcal{L}_{KL}}. \tag{50}$$

The reconstruction term can easily be computed in a variational auto-encoder framework. Below, we provide the derivation of of the KL term:

$$\mathcal{L}_{KL} = \mathbb{E}_{q_\phi(\mathbf{z}_{0:T},\mathbf{s}_{1:T})} \left[ \log \frac{p_\theta(\mathbf{z}_0)}{q_\phi(\mathbf{z}_0|\mathbf{x}_{1:T},\mathbf{u})} + \log \frac{p_\theta(\mathbf{z}_{1:T},\mathbf{s}_{1:T}|\mathbf{u})}{q_\phi(\mathbf{z}_{1:T},\mathbf{s}_{1:T}|\mathbf{x}_{1:T},\mathbf{u})} \right] \tag{51}$$

$$= -D_{KL}(q_\phi(\mathbf{z}_0|\mathbf{x}_{1:T},\mathbf{u})||p_\theta(\mathbf{z}_0)) + \mathbb{E}_{q_\phi(\mathbf{z}_{0:T},\mathbf{s}_{1:T})} \left[ \log \frac{p_\theta(\mathbf{z}_{1:T},\mathbf{s}_{1:T}|\mathbf{u})}{q_\phi(\mathbf{z}_{1:T},\mathbf{s}_{1:T}|\mathbf{x}_{1:T},\mathbf{u})} \right] \tag{52}$$

$$= -D_{KL}(q_\phi(\mathbf{z}_0|\mathbf{x}_{1:T},\mathbf{u})||p_\theta(\mathbf{z}_0)) + \mathbb{E}_{q_\phi(\mathbf{z}_{0:T},\mathbf{s}_{1:T})} \left[ \sum_{t=1}^{T} \log \frac{p_\theta(\mathbf{s}_t|\mathbf{u})p_\theta(\mathbf{z}_t|\mathbf{z}_{t-1},\mathbf{s}_t)}{q_\phi(\mathbf{s}_t|\mathbf{z}_{t-1},\mathbf{x}_{1:T},\mathbf{u})q_\phi(\mathbf{z}_t|\mathbf{z}_{t-1},\mathbf{s}_t)} \right] \tag{53}$$

$$= -D_{KL}(q_\phi(\mathbf{z}_0|\mathbf{x}_{1:T},\mathbf{u})||p_\theta(\mathbf{z}_0)) + \sum_{t=1}^{T} \mathbb{E}_{q_\phi(\mathbf{z}_{0:T},\mathbf{s}_{1:T})} \left[ \log \frac{p_\theta(\mathbf{s}_t|\mathbf{u})}{q_\phi(\mathbf{s}_t|\mathbf{z}_{t-1},\mathbf{x}_{1:T},\mathbf{u})} \right] \tag{54}$$

$$= -D_{KL}(q_\phi(\mathbf{z}_0|\mathbf{x}_{1:T},\mathbf{u})||p_\theta(\mathbf{z}_0)) + \sum_{t=1}^{T} \underbrace{\mathbb{E}_{q_\phi(\mathbf{s}_t,\mathbf{z}_{t-1}|\mathbf{z}_{t-2},\mathbf{s}_{t-1},\mathbf{x}_{1:T},\mathbf{u})} \left[ \log \frac{p_\theta(\mathbf{s}_t|\mathbf{u})}{q_\phi(\mathbf{s}_t|\mathbf{z}_{t-1},\mathbf{x}_{1:T},\mathbf{u})} \right]}_{-\mathbb{E}_{q_\phi(\mathbf{z}_{t-1}|\mathbf{z}_{t-2},\mathbf{s}_{t-1})}[D_{KL}(q_\phi(\mathbf{s}_t|\mathbf{z}_{t-1},\mathbf{x}_{1:T},\mathbf{u})||p_\theta(\mathbf{s}_t|\mathbf{u}))]} \tag{55}$$

# E   ADDITIONAL EXPERIMENTAL DISCUSSION AND RESULTS

## E.1   IS NOISE HARDER TO ESTIMATE?

In Sec. 4, we observe that $\text{MCC}[\bar{\mathbf{z}}_{\text{input}}] > \text{MCC}[\bar{\mathbf{s}}_{\text{input}}]$ for all models. In the generative model, observations $\mathbf{x}(t)$ directly depend on $\mathbf{z}(t)$: $\mathbf{x}(t) = \mathbf{g}(\mathbf{z}(t))$, while the dependency to $\mathbf{s}(t)$ is indirectly through $\mathbf{z}(t)$: $\mathbf{x}(t) = \mathbf{g}(\mathbf{f}(\mathbf{z}_{t-1},\mathbf{s}_t))$. To reconstruct observations $\mathbf{x}(t)$, the model has to recover $\mathbf{z}(t)$ without having to fully identify the noise $\mathbf{s}(t)$. Hence, the noise is harder to estimate, to the best of our understanding.

## E.2   ABLATION: HOW DOES EACH MODEL COMPONENT AFFECT THE PERFORMANCE?

To understand how each model component affects the performance, we set up an ablation study. We chose $\beta$-VAE (Higgins et al., 2017) as the base model. On top of it, we add three model components one by one: (i) 1-MLP-DYN: a single MLP modeling $K$ outputs of the transition function $\{f_k\}_{k=1}^{K}$, (ii) $K$-MLP-DYN: $K$ independent MLPs modeling $K$ outputs of the transition function $\{f_k\}_{k=1}^{K}$, and (iii) OURS: an additional conditional normalizing flow to model the nonstationarity $p(\mathbf{s}|\mathbf{u})$. We present $\text{MSE}[\bar{\mathbf{x}}_{\text{future}}]$ results for prediction steps $T_{\text{pred}} \in \{2,4,8\}$ in Fig. 7, together with the $\text{MCC}[\bar{\mathbf{z}}_{\text{input}}]$ values in the legend. $\beta$-VAE cannot make future predictions, hence we only report its $\text{MCC}[\bar{\mathbf{z}}_{\text{input}}]$ result. We see that adding $K-\text{MLP}$ transitions results in larger performance improvements both in terms of disentanglement ($\text{MCC}[\bar{\mathbf{z}}_{\text{input}}]$) and future predictions ($\text{MSE}[\bar{\mathbf{x}}_{\text{future}}]$), compared to adding $1-\text{MLP}$ transitions or

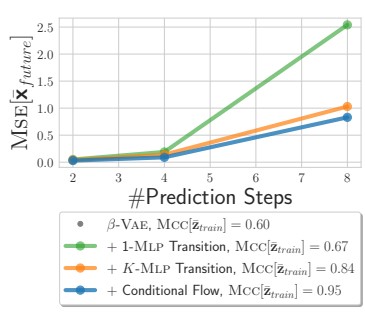

Figure 7: MCC vs. MSE results for ablations for different time steps.

conditional flows. As expected, the improvement in future prediction accuracy is larger compared to the improvement in disentanglement performance, since conditional flows only indirectly affect the future predictions through the noise samples while directly affecting the noise distribution. More importantly, the results suggest that as we predict for longer horizons (e.g. 8 steps), a higher MCC on the latent states corresponds to a larger improvement on the future prediction performance.

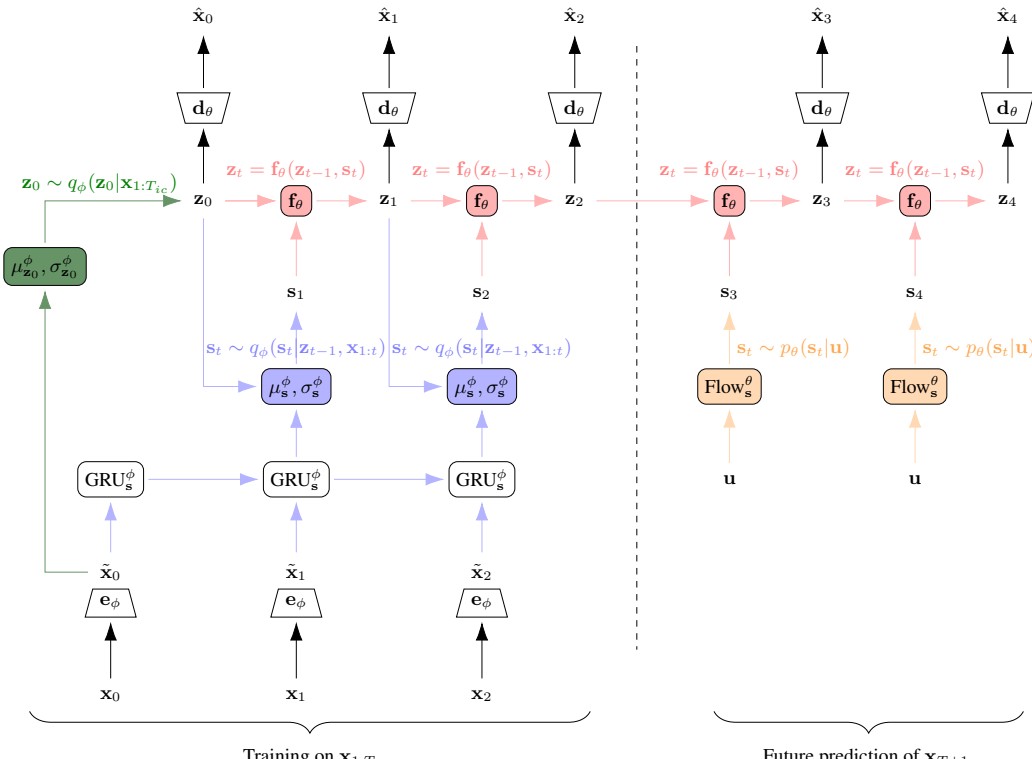

Figure 6: Diagram of the model architecture: In training, the observation $\mathbf{x}$ is passed through the encoder $\mathbf{e}_\phi$ to get the intermediate representation $\tilde{\mathbf{x}}$. We learn the distribution over the initial latent state $\mathbf{z}_0$ conditional on the representations of the first $T_{ic}$ observations (in the diagram, we show $T_{ic} = 1$; in our experiments, we use $T_{\text{ic}} = 2$). The latent state is decoded by the decoder $\mathbf{d}_\theta$ to produce the predicted observation $\hat{\mathbf{x}}$ (which is trained to match the corresponding actual observation). The next value of the latent state is computed by the transition function $\mathbf{f}_\theta$, which depends both on the previous state and on the process noise $\mathbf{s}$. In training, the process noise $\mathbf{s}$ is sampled from the variational posterior that depends on the previous state as well as on the representation created by a recurrent neural network (GRU) that has received up to the current observation. In future prediction, the process noise $\mathbf{s}$ is sampled from the prior, which is a learned normalizing flow.

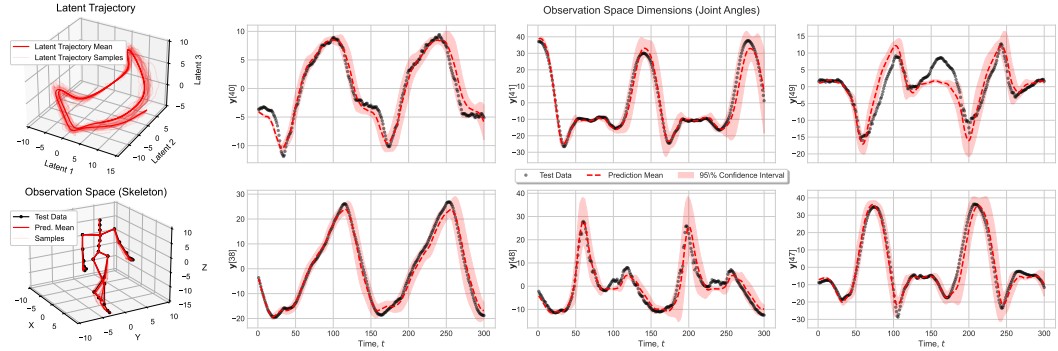

Figure 8: Test trajectory prediction samples, their corresponding skeletons and latent trajectories for MOCAP-SINGLE. At test time, the model takes first 3 time points as input, and predicts the full horizon ($T = 300$).

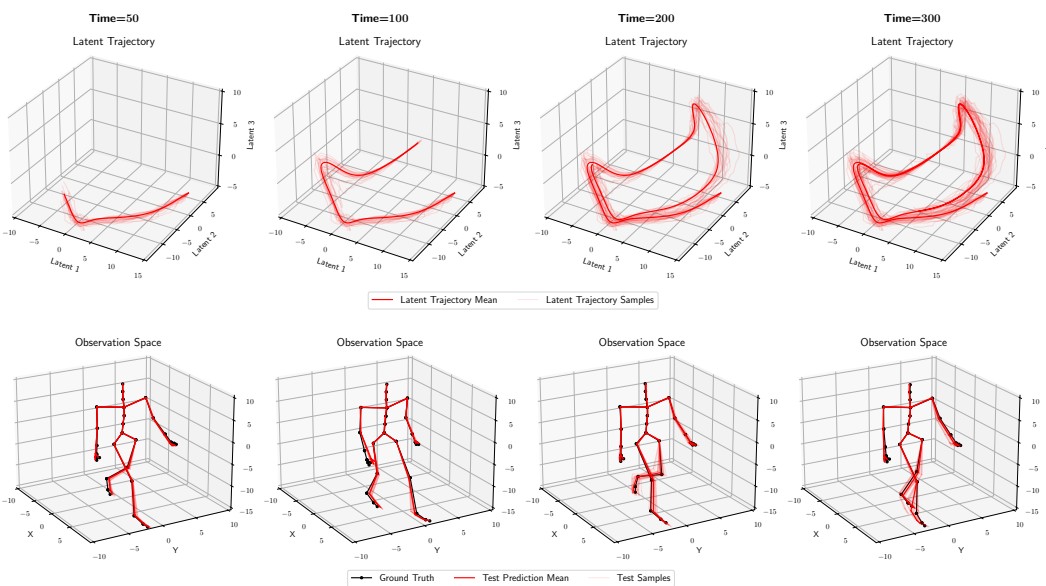

Figure 9: Test predictions for MOCAP-SINGLE. **(Top row)** From left to right, each column shows latent trajectory samples until times $[50, 100, 200, 300]$. **(Bottom row)** From left to right, each column shows samples at times $[50, 100, 200, 300]$ transformed into skeletons, together with test data.

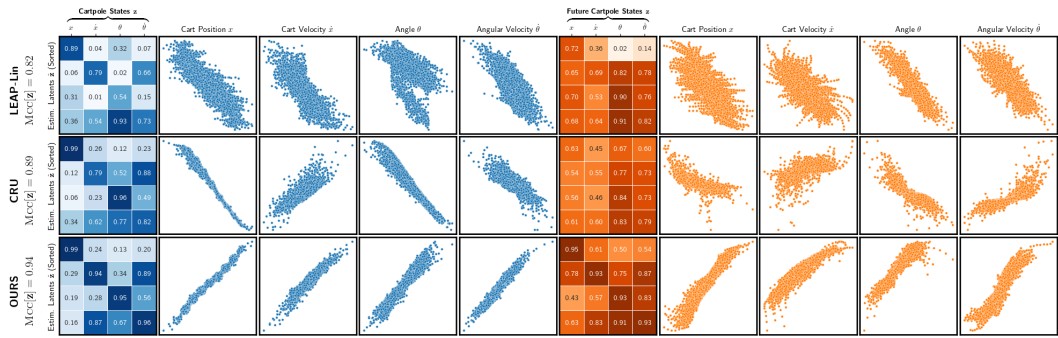

Figure 10: Extension of Fig. 2. **(Left 4 columns)** The same columns as Fig. 2. **(Right 4 columns)** The same plots produced for the 8-step future predictions.

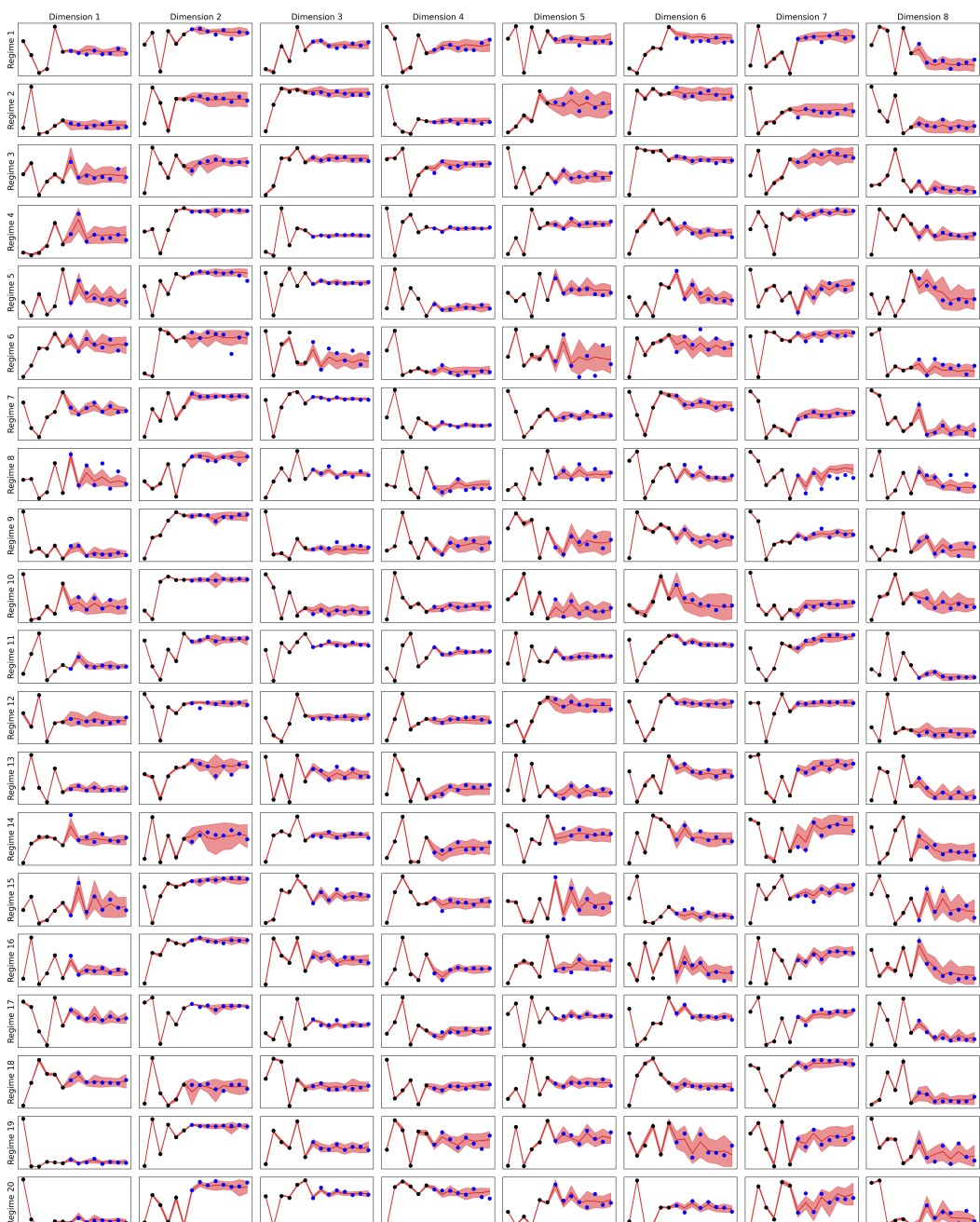

Figure 11: For the synthetic experiment, model predictions in the data space with the estimated uncertainties. Our model uses the first $T_0 = 2$ data points to encode the initial latent state $\mathbf{z}_0$ and $T_{\text{dyn}} = 4$ data points to encode the noise sequence $\bar{\mathbf{s}}_{\text{train}}$. We unroll our model for $T_0 + T_{\text{dyn}} + T_{\text{future}} = 14$ steps ahead, where the future noise variables $\mathbf{s}_{7:14}$ follow the learned prior flow. We draw 32 trajectory samples $\mathbf{x}_{0:14}$ by sampling from the initial state and process noise. Above, black and blue dots show the training and test data points. The red curves are the mean trajectories and the red region corresponds to $\pm 2$ standard deviation computed empirically. We observe near-perfect predictions and low uncertainty for the input data (the first $T_{\text{train}} = T_0 + T_{\text{dyn}} = 6$ time points) while the uncertainty grows as we unroll over time. Further, the uncertainty grows even more when the model predictions are off. Therefore, almost all test points lie in the $\pm 2$ std region, reflecting the high calibration level our probabilistic model attains.

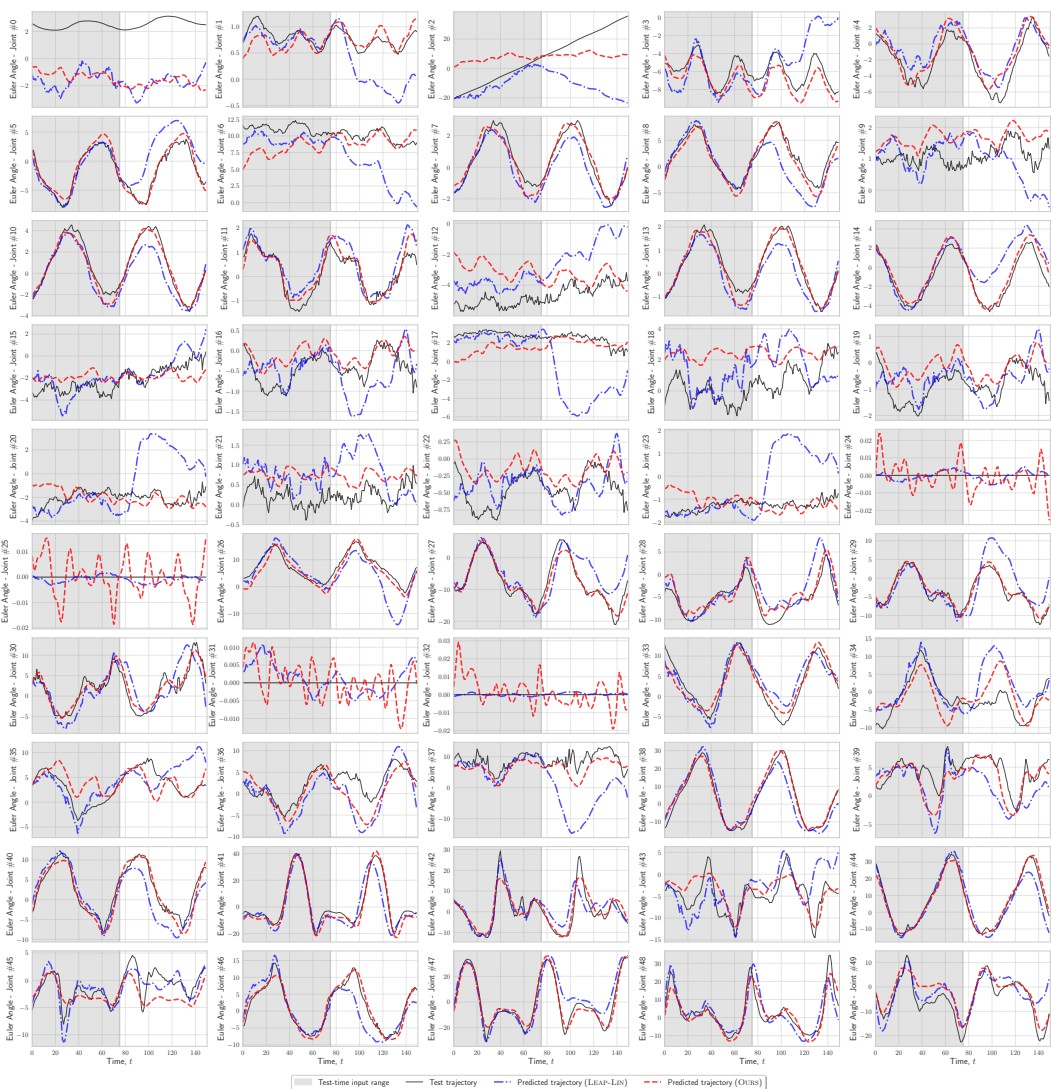

Figure 12: Test trajectory predictions for all 50 dimensions for LEAP-LIN and OURS in MOCAP-MULTI. At test time, first 75 time points are given as input, and the full prediction horizon is $T = 150$ time points.

## F EXPERIMENTAL DETAILS

### F.1 DATASETS

We chose three datasets with different characteristics to keep our experimental setup (i) as general as possible, and (ii) comparable with the related works that are closest to ours (LEAP (Yao et al., 2021) and TDRL (Yao et al., 2022)). Our datasets include: (i) a synthetic dataset satisfying our modeling assumptions as common in nonlinear ICA literature (Hyvärinen & Morioka, 2016; 2017; Hyvärinen et al., 2019; Yao et al., 2021; 2022), (ii) a cartpole dataset with high-dimensional video observations whose low-dimensional ground-truth dynamics are governed by a set of nonlinear differential equations (Yao et al., 2022), and (iii) the high-dimensional (50-dim) real-world Mocap dataset with longer training and prediction horizons (ranging between 75-300) (Yao et al., 2021; 2022; Yildiz et al., 2019; Li et al., 2020; Auzina et al., 2024).

Our motivation for the synthetic experiment is to validate our theory, and for the cartpole experiment is to show our model's ability to recover low-dimensional latent dynamics from a different kind of

high-dimensional input modality (videos). To test our model's capability to predict longer sequences of complex real-world dynamics, we also included the MOCAP experiment.

**Note on nonstationarity and observation regimes.** In the experiments, we observe $(\mathbf{x}_t, \mathbf{u}_t)$ pairs where each observation $\mathbf{x}_t$ has an associated regime label $\mathbf{u}_t$. For the synthetic experiment, we have $R = 20$ regimes. For the cartpole experiment, we have $R = 6$ regimes (5 for training, 1 for testing). For the MOCAP experiment, the different regimes correspond to different persons walking. In MOCAP-SINGLE, we have just one person, hence $R = 1$, and $\mathbf{u}_t$ is constant, demonstrating a case violating our assumptions. In MOCAP-MULTI and MOCAP-SHIFT, we have 6 persons, hence $R = 6$. For each observation $\mathbf{x}_t$, the model (or the prior $p(\mathbf{s}_t|\mathbf{u})$) takes $\mathbf{u}_t$, i.e., the one-hot encoded regime label, as input.

**Synthetic dataset.** We first set up a synthetic experiment to evaluate latent dynamics identification. Same as Yao et al. (2021; 2022), we generate multivariate time-series following our data generating process in Eqs. (1)–(4): the dimension of $\mathbf{s}$, $\mathbf{z}$ and $\mathbf{x}$ is $K = 8$, the number of regimes is $R = 20$, the distribution of noise $s$ changes between distinct regimes to provide non-stationarity to satisfy our assumptions, $s_{kt} = \frac{1}{a_k(\mathbf{u})}\epsilon_t, \epsilon \sim \mathcal{N}(0, 0.1)$, the functions $\mathbf{f}$ and $\mathbf{g}$ are a 2-linear layer random MLPs, and we choose $\mathbf{z}_{1:T}$ to be a second-order Markov process, i.e., $\mathbf{z}_t = \mathbf{f}(\mathbf{z}_{t-2}, \mathbf{z}_{t-1}, \mathbf{s}_t)$. We summarize the generative process as follows:

$$\mathbf{z}_0, \mathbf{z}_1 \sim \mathcal{N}(0, 1), \tag{56}$$

$$s_{kt} = \frac{1}{a_k(\mathbf{u})}\epsilon_t, \qquad \epsilon_t \sim \mathcal{N}(0, 0.1), \qquad t = 2, ..., T, \tag{57}$$

$$\mathbf{z}_t = \mathbf{f}(\mathbf{z}_{t-2}, \mathbf{z}_{t-1}, \mathbf{s}_t), \qquad t = 2, ..., T, \tag{58}$$

$$\mathbf{x}_t = \mathbf{g}(\mathbf{z}_t), \qquad t = 0, ..., T, \tag{59}$$

where $a_k(\mathbf{u}) \sim U(0, 1)$ is a uniform random variable that modulates the noise variance between distinct regimes. Hence, the noise $\mathbf{s}$ represents random or unmodeled external variations in the process, not captured by deterministic dynamics. Here, the distribution of $\mathbf{s}$ simply changes between distinct regimes to provide non-stationarity to satisfy the identifiability assumptions (see App. F.2).

For each environment, we generate $7500/750/750$ sequences as train/validation/test data. We chose the synthetic training sequences as $T_0 + T_{\text{dyn}} = 2 + 4 = 6$ as in LEAP (Yao et al., 2021) and TDRL (Yao et al., 2022), to keep our setup comparable to the two closest related works. Including the future time points, each sequence has the total length $T = T_0 + T_{\text{dyn}} + T_{\text{future}} = 2 + 4 + 8 = 14$. As we have a second-order Markov process, first $T_0 = 2$ states ($\mathbf{z}_0, \mathbf{z}_1$) are spared as initial states. The next 4 observations $\mathbf{x}_{2:5}$ are used for training the dynamical model. The last 8 observations $\mathbf{x}_{6:13}$ are used for evaluating future predictions. We choose the future prediction horizon $T_{\text{future}} = 8$ as the double of the training sequence length. If the dynamics are truly identified, the model should predict future states well, even for this longer horizon.

**Cartpole dataset.** Next, we use a more challenging video dataset of a cartpole system (Brockman et al., 2016). The environment simulates a latent physical system with a pole attached to a cart from a pivot point. The system has 4 states: position and velocity of the cart, and the angle and the angular velocity of the pole ($[x_{\text{cart}}, \theta_{\text{pole}}, \dot{x}_{\text{cart}}, \dot{\theta}_{\text{pole}}]$). Observations consist of video sequences of the system and we use the true states only to compute MCC.

As in Huang et al. (2021); Yao et al. (2022), we use a modified setup with: (i) 5 source domains used for training with different levels of gravity $g = \{5, 10, 20, 30, 40\}$ and mass $m = 1.0$, and (ii) a target domain with $g = 90$ to check the methods' ability to *extrapolate*. The regime labels are non-informative and observed, while the gravity values are unobserved. However, Yao et al. (2022) assign a single constant action (only left or only right) to each environment, which is very unrealistic. In contrast to this, we observe the system under random binary actions, which move the cart to left or right at each step. We generate sequences of length $T = T_0 + T_{\text{dyn}} + T_{\text{future}} = 1 + 7 + 8 = 16$: the first frame for the initial state, the next 7 frames for training the dynamical model, and the last 8 frames for future prediction. For each source domain, we have $900/100/100$ sequences as train/validation/test data. For the target domain, we have data sets with different numbers of samples $N_{\text{target}} = \{20, 50, 100, 1000\}$.

**Mocap datasets.** To evaluate future predictions on long horizons with complex real-world dynamics, we use three datasets from the CMU motion capture (Mocap) library containing human action trajectories (Wang et al., 2007; Yao et al., 2021; 2022): MOCAP-SINGLE (Yildiz et al., 2019), MOCAP-MULTI (Auzina et al., 2024) and MOCAP-SHIFT (Auzina et al., 2024). For all Mocap datasets, each observation $\mathbf{x}_t \in \mathbb{R}^{50}$ has 50 dimensions, after the preprocessing used in (Yildiz et al., 2019; Li et al., 2020).

MOCAP-SINGLE contains 23 walking trials of single subject 35, split into 16/3/4 train/val/test sequences. Each sequence has 300 time points. At test time, the models take first 3 data points as input and predict the whole trajectory. MOCAP-MULTI and MOCAP-SHIFT contain a total of 56 walking trials for 6 subjects. Each sequence is down-sampled by a factor of 2 to 150 time points. The models take first 75 data points as input and predict the whole trajectory. The detailed subject and trial ids can be found in the Appendix of Auzina et al. (2024).

### F.2 ASSUMPTIONS (A3, A4) ON SYNTHETIC AND CARTPOLE DATASETS

In the synthetic experiment, we use noise with (randomly) modulated variance between environments. Under this modulated noise, sufficient variability assumptions *(A3, A4)* hold.

**Assumption *(A4)* in the synthetic experiment.** We start with the analysis for the noise $\mathbf{s}$ (assumption *(A4)*). Our ground-truth functions for this experiment are:

$$s_{kt} = \frac{1}{a_k(\mathbf{u})}\epsilon_t, \tag{60}$$

$$z_{kt} = f_k(\mathbf{z}_{t-1}, \mathbf{z}_{t-2}) + s_{kt}. \tag{61}$$

Without loss of generality, let us assume $\epsilon$ is a standard normal random variable: $\epsilon \sim N(0,1)$ to simplify the derivations. The log probability of noise is:

$$\log p(s_{kt}|\mathbf{u}) = q_{kt} = -\log 2\pi + \log a_k - \frac{a_k^2}{2}s_{kt}^2.$$

Then, the required partial derivatives take the following form:

$$\frac{\partial^2 q_{kt}}{\partial s_{kt}\partial u_l} = -2a_k(\mathbf{u})\frac{\partial a_k}{\partial u_l}(\mathbf{u})s_{kt}, \tag{62}$$

$$\frac{\partial^3 q_{kt}}{\partial s_{kt}^2\partial u_l} = -2a_k(\mathbf{u})\frac{\partial a_k}{\partial u_l}(\mathbf{u}). \tag{63}$$

These two equations tell us that if the $2K$ functions $-2a_k(\mathbf{u})\frac{\partial a_k}{\partial u_l}(\mathbf{u})s_{kt}$ and $-a_k(\mathbf{u})\frac{\partial a_k}{\partial u_l}(\mathbf{u})$ for $k = 1, \ldots, K$ in $\mathbf{u}$ are linearly independent, then the assumption *(A4)* holds.

For our synthetic setup, we choose the modulator variables $a_k(\mathbf{u})$ uniformly random between $[0,1]$. To simplify the analysis, let us assume that we have $a_k = \sqrt{2u_l}w_k$ with $w_k \sim U(0,1)$. Then, the $2K$ functions become $-2w_k s_{kt}$ and $-w_k$, which are random functions. This results in a $2K \times 2K$ matrix with random entries in the form of $-w_k$ and $-2w_k s_{kt}$ with $w_k \sim U(0,1)$ and $s_{kt} \sim \mathcal{N}(0, a_k^{-2})$. We conjecture that the probability of these random vectors to be linearly independent is 1. To validate this empirically, we sampled 10000 $2K \times 2K$ matrices with entries $-2w_k s_{kt}$ and $-w_k$, where $w_k \sim U(0,1)$ and $s_{kt} \sim N(0, w_k^2)$. They all had rank $2K$.

**Assumption *(A3)* in the synthetic experiment.** The analysis for the latent $\mathbf{z}$ (assumption *(A3)*) is similar. We write the log probability as follows:

$$\log p(z_{kt}|\mathbf{z}_{t-1},\mathbf{u}) = \eta_{kt} = -\log 2\pi + \log a_k - \frac{a_k^2}{2}(z_{kt} - f_k(\mathbf{z}_{t-1}))^2.$$

Then, the required partial derivatives take the following form:

$$\frac{\partial^2 \eta_{kt}}{\partial z_{kt}\partial u_l} = -2a_k(\mathbf{u})\frac{\partial a_k}{\partial u_l}(\mathbf{u})(z_{kt} - f_k(\mathbf{z}_{t-1})), \tag{64}$$

$$\frac{\partial^3 \eta_{kt}}{\partial z_{kt}^2\partial u_l} = -2a_k(\mathbf{u})\frac{\partial a_k}{\partial u_l}(\mathbf{u}). \tag{65}$$

Since $z_{kt} - f_k(\mathbf{z}_{t-1}) = s_{kt}$, the $2K$ equations here have the same form as in Eqs. (62) and (63) in the analysis of assumption *(A4)*. Then, using the same reasoning, the assumption *(A3)* holds.

Table 4: Architecture components for the synthetic experiment ($T_{\text{input}} = 2 + 4 = 6, T_{\text{ic}} = 4, D = 8, L = 2$).

| Base Encoder | ICEncoder | NoiseEncoder | TransitionMLP ($D\times$ MLPs) | Decoder |
|---|---|---|---|---|
| Input: $\mathbb{R}^{T_{\text{input}} \times D}$ | Input: $\mathbb{R}^{T_{\text{ic}} \times 64}$ | Input: $\mathbb{R}^{T_{\text{input}} \times 64}$ | $D\times$ Input: $\mathbb{R}^{L \times D+1}$ | Input: $\mathbb{R}^{T_{\text{input}} \times D}$ |
| FC 64, Leaky-ReLU | FC 64, Leaky-ReLU | GRU 64, 1 layer | $D\times$ FC 64, Leaky-ReLU | FC 64, Leaky-ReLU |
| FC 64, Leaky-ReLU | FC 64, Leaky-ReLU | Concat[$\cdot, \mathbf{z}_{t-L:t-1}$] | $D\times$ FC 64, Leaky-ReLU | FC 64, Leaky-ReLU |
| FC 64, Leaky-ReLU | FC $L \times 2 \times D$ | FC 64, Leaky-ReLU | $D\times$ FC 1 | FC 8 |
| FC 64 | | FC 64, Leaky-ReLU | | |
| | | FC $2 \times D$ | | |

Table 5: Other hyperparameters for our method for the synthetic experiment.

| Parameter | Value |
|---|---|
| lr | 2e-3 |
| $\beta$ | 2.5e-03 |
| $\beta_{\mathbf{z}_0}$ | 1e-03 |
| weight decay | 1e-6 |
| batch size | 32 |
| latent dim. | 8 |

**Cartpole dataset.** While we simulate cartpole data under different gravity values to create nonstationarity, Assumptions *(A3, A4)* do not exactly hold in the cartpole experiment since the nonlinear differential equations underlying the cartpole system are deterministic.

### F.3 ARCHITECTURE AND OPTIMIZATION DETAILS

In this section, we report the architecture details. We optimize our model with Adam optimizer with default parameters, except the learning rate which is chosen by validation. We chose all hyperparameters for our method, KALMANVAE, two versions of LEAP, CRU and TDRL with cross-validation. $\beta$-VAE uses the hyperparameters chosen for our method. In particular, we performed random search as well as Bayesian optimization (in some cases) over learning rate, loss weights (e.g., $\beta$), weight regularization, the number of layers in all MLPs, and latent dimensionality.

**Conditional normalizing flows.** In all experiments, we use 1D conditional normalizing flows to model the nonstationary prior for the noise variables. We employ the implementation from Stimper et al. (2023). These flows are 1-layer neural spline flows conditioned on the auxiliary variable $\mathbf{u}$. The auxiliary variable $\mathbf{u}$ is first one-hot encoded and then embedded using 1 linear layer before taken as input.

#### F.3.1 SYNTHETIC DATA EXPERIMENTS

**Our method.** For the synthetic experiment, the architectural components of our model are detailed in Tab. 4. In addition to the architecture layers, our method has several other hyperparameters selected by validation. They are listed in Tab. 5.

**Baseline methods.** For the baselines KALMANVAE, CRU, TDRL, LEAP-LIN and LEAP-NP, we use the architectures provided for multivariate time-series in their public code base. The identifiable representation learning methods TDRL, LEAP-LIN and LEAP-NP have similar architectures to ours, since they also perform synthetic experiments to validate their theoretical findings. The rest of the hyperparameters are specified in Tab. 6. For the hyperparameters with multiple values, we perform a sweep over them to determine the final value by validation. For the others with a single value, we keep the recommended values in the publicly available code base.

#### F.3.2 CARTPOLE EXPERIMENTS

For the cartpole experiment, the architectural components of our model used in the cartpole experiment are detailed in Tab. 7. We present the rest of the hyperparameters for our method in Tab. 8.

Table 6: Hyperparameters of the baseline methods for the synthetic experiment.

| Model | Parameter | Value |
|---|---|---|
| TDRL | learning rate | [2e-3, 1e-3] |
| | $\beta$ | [1e-2, 3e-3, 2e-3, 1e-3, 1e-4] |
| | $\gamma$ | 1e-2 |
| | latent dim. | 8 |
| LEAP-LIN and LEAP-NP | learning rate | [2e-3, 1e-3] |
| | $\beta$ | [1e-2, 3e-3, 2e-3, 1e-3, 1e-4] |
| | $\gamma$ | 2e-2 (LEAP-LIN), 9e-3 (LEAP-NP) |
| | $\sigma$ | 1e-6 |
| | latent dim. | 8 |
| CRU | learning rate | [5e-3, 3e-3, 1e-3] |
| | latent dim. | [16, 32] |
| KALMANVAE | learning rate | 1e-3 |
| | $\beta$ | [10, 1, 0.1, 1e-2, 1e-3] |
| | init. cov | [10, 1, 0.1] |
| | latent dim. | [2, 4, 8] |

Table 7: Architecture components for the cartpole experiment ($T_{\text{input}} = 1 + 7 = 8, T_{\text{ic}} = 4, L = 1, K = 8$).

| Encoder CNN | Decoder Deconv | |
|---|---|---|
| Input: $T_{\text{input}} \times 64 \times 64 \times$ nc | Input: $T_{\text{input}} \times \mathbb{R}^K$ | |
| $3 \times 3$ Conv2d, 32 GeLU, stride 2 | FC 64, Unflatten | |
| $3 \times 3$ Conv2d, 32 GeLU, stride 2 | $3 \times 3$ Upconv2d, 64 GeLU, stride 2 | |
| $3 \times 3$ Conv2d, 64 GeLU, stride 2 | $3 \times 3$ Upconv2d, 32 GeLU, stride 2 | |
| $3 \times 3$ Conv2d, 64 GeLU, stride 2 | $3 \times 3$ Upconv2d, 32 GeLU, stride 2 | |
| Flatten | $3 \times 3$ Upconv2d, nc Sigmoid, stride 2 | |

| ICEncoder | NoiseEncoder | TransitionMLP ($K \times$ MLPs) |
|---|---|---|
| Input: $\mathbb{R}^{T_{\text{ic}} \times 64}$ | Input: $\mathbb{R}^{T_{\text{input}} \times 64}$ | $D \times$ Input: $\mathbb{R}^{L \times D + 1}$ |
| FC 64, Leaky-ReLU | GRU 64, 1 layer | $D \times$ FC 64, Leaky-ReLU |
| FC 64, Leaky-ReLU | Concat$[\cdot, \mathbf{z}_{t-L:t-1}]$ | $D \times$ FC 64, Leaky-ReLU |
| FC $L \times 2 \times D$ | FC 64, Leaky-ReLU | $D \times$ FC 1 |
| | FC 64, Leaky-ReLU | |
| | FC $2 \times D$ | |

Table 8: Other hyperparameters for our method for the cartpole experiment.

| Name | Value |
|---|---|
| lr | 2e-4 |
| $\beta$ | 1e-1 |
| $\beta_{\mathbf{z}_0}$ | 1e-1 |
| weight decay | 0.0 |
| batch size | 8 |
| latent dim. | 8 |

**Baseline methods.** Among the baselines, TDRL (Yao et al., 2022) also has an experiment on the modified cartpole setup. We keep their architecture, including the backbone CNN and its deconvolution decoder. For CRU, LEAP-LIN (Yao et al., 2021) and LEAP-NP (Yao et al., 2021), we use the same backbone CNN and the deconvolutional decoder from Yao et al. (2022), since it is a

larger network compared to ours with 1 additional conv2d layer (filters: 32,32,64,64,64), i.e., it has more parameters. For KALMANVAE, we tried the CNN encoder/decoder components from TDRL Yao et al. (2022) and our architecture, but the algorithm kept throwing numerical errors among 5 seeds (it finished only 1/5 seed without errors). Therefore, we used the simple CNN encoder used in their own implementation. The rest of the hyperparameters are specified in Tab. 9.

Table 9: Hyperparameters of the baseline methods for the cartpole experiment.

| Model | Parameter | Value |
|---|---|---|
| TDRL | learning rate | [2e-3, 1e-3] |
| | $\beta$ | [1.0, 1e-1, 1e-2, 1e-3] |
| | $\gamma$ | [100, 10, 1] |
| | $\sigma$ | [1e-2, 1e-3] |
| | latent dim. | 8 |
| LEAP-LIN and LEAP-NP | learning rate | [2e-3, 1e-3, 5e-4] |
| | $\beta$ | [1.0, 1e-1, 1e-2] |
| | $\gamma$ | [20, 10, 1] |
| | $\sigma$ | [1e-6, 0.0] |
| | $\lambda_{l_1}$ | [1e-1, 1e-2, 0.0] |
| | latent dim. | 8 |
| CRU | learning rate | [5e-3, 3e-3, 1e-3] |
| | latent dim. | [16, 32] |
| KALMANVAE | learning rate | [7e-3, 3e-3, 1e-3] |
| | $\beta$ | [1, 0.1] |
| | init. cov | [20, 1, 0.1] |
| | latent dim. | [2, 4, 8] |

### F.3.3 MOCAP EXPERIMENTS

For Mocap experiments, we adapt the architecture used in Yildiz et al. (2019) to our model, e.g., we use the same number of layers and hidden dimensions for the base encoder and the decoder. We also use its observational density which is a diagonal Gaussian where a single diagonal variance parameter is estimated. The details are presented in Tab. 10.

Table 10: Architecture components for the Mocap experiments ($T_{ic} = 3, D = 50, K = 3, L = 1$).

| Base Encoder | ICEncoder | NoiseEncoder | TransitionMLP | Decoder |
|---|---|---|---|---|
| Input: $\mathbb{R}^{T_{input} \times D}$ | Input: $\mathbb{R}^{T_{ic} \times 30}$ | Input: $\mathbb{R}^{T_{input} \times 30}$ | $D \times$ Input: $\mathbb{R}^{L \times D + 1}$ | Input: $\mathbb{R}^{T_{input} \times D}$ |
| FC 30, Leaky-ReLU | FC 30, Leaky-ReLU | GRU 30, 1 layer | FC 30, Leaky-ReLU | FC 30, Leaky-ReLU |
| FC 30 | FC $L \times 2 \times K$ | Concat$[\cdot, \mathbf{z}_{t-L:t-1}]$ | FC 30, Leaky-ReLU | FC 30, Leaky-ReLU |
| | | FC 30, Leaky-ReLU | FC $K$ | FC $D + 1$ |
| | | FC $2 \times D$ | | |

The rest of the hyperparameters for our method are shown in Tab. 11. Additionally, we have observed that gradually increasing the training sequence length during training improves training stability for our method. For example, we start training with shorter sequences, e.g., with 25-step long and we train the model with these for 1 epoch. Afterwards, the training continues with 75-step long sequences until convergence.

Table 11: Other hyperparameters for our method for the Mocap experiment.

| Name | Value |
|---|---|
| lr | 2e-4 |
| $\beta$ | 1e-1 |
| $\beta_{\mathbf{z}_0}$ | 1e-1 |
| weight decay | 0.0 |
| batch size | 8 |
| latent dim. | 3 |

**Baseline methods.** For LEAP-LIN (Yao et al., 2021), we use its recommended architecture for Mocap datasets, since it also has an experiment on another Mocap dataset. For CRU, we adapt the encoder and decoder layers in Yildiz et al. (2019). The rest of the hyperparameters are specified in Tab. 12. Particularly, we tried to increase the input sequence length, but we observed that the model cannot be trained with sequences longer than 10-steps.

Table 12: Hyperparameters of the baseline methods for the Mocap experiment.

| Model | Parameter | Value |
|---|---|---|
| CRU | learning rate | [5e-3, 3e-3, 1e-3] |
| | latent dim. | [16, 32] |
| LEAP-LIN | learning rate | [1e-3, 5e-4, 1e-4] |
| | $\beta$ | [1.0, 1e-1] |
| | $\gamma$ | [20, 10, 1] |
| | $\sigma$ | [1e-6, 0.0] |
| | $\lambda_{l_1}$ | [1e-3, 0.0] |
| | latent dim. | 8 |
| | sequence length | [5,7,10] |

