# OpenReview forum: "Identifying latent state transitions in non-linear dynamical systems"
_ICLR.cc/2025/Conference — ICLR 2025 Poster_

### Official Review · Reviewer_oxFU · 2024-10-28

**Soundness:** 3
**Presentation:** 3
**Contribution:** 2
**Rating:** 6
**Confidence:** 3

**Summary:**

This paper presents a framework for identifying the unknown transition function in latent dynamical systems. The authors emphasize that previous research focused on identifying latent states, and lack the ability to accurately predict future states, particularly in complex nonlinear systems.

The authors establish the theoretical identifiability of both the latent states and the transition function. They leverage the assumption of "sufficient variability," meaning that the latent noise and states exhibit enough variation across different environments to disentangle the underlying dynamics. The framework also relies on the assumption that the augmented transition function is bijective.

The framework was tested across a range of scenarios, including synthetic datasets, a simulated cartpole environment, and real-world Mocap data. The results demonstrate that the model effectively recovers the true underlying dynamics, and is accurate in predicting future states.

After the discussion, the authors have addressed my concerns. I have decided to raise the score.

**Strengths:**

The identifiability was theoretically defined and justified with given assumptions.

The proposed framework was empirically evaluated and compared with existing methods across synthetic and real-world data.

**Weaknesses:**

The claimed contribution "the first framework that allows for the identification of the unknown transition function alongside latent states and the generative function" seems overstated. There are existing methods to identify latent states, transition and generative functions together, e.g. [Sussillo16, Schimel21, Hess23, Zhao23].

The proposed framework is not demonstrated as general as claimed like "... improves generalization and interpretability of dynamical systems...". The authors focused on typical RL and robotics scenario. There exist dynamical systems in other fields do not satisfy those assumptions or are in completely different regime. This could disadvantage the compared methods with weaker assumptions (e.g. NODE).
Though the authors used Mocap dataset to showcase then it violates the identifiability assumptions, the system itself is still of RL and robotics scenario. It is totally valid to narrow down problems to solve them better. However, the statements and claims should be more specific.

**Questions:**

Does the proposed framework requires the dimensionality of latent space and noise to be known?
If not, what would be the procedure of estimation?

Does the proposed framework allow for sparse noise?

---

> ### Author Response · Authors · 2024-11-21
>
> ***W1) Confusion around the term “identification”:***
>
> There seems to be some confusion around the term “identification” due to its different meanings in (i) dynamical systems and (ii) nonlinear ICA literature. In the mentioned works [Sussillo16, Schimel21, Hess23, Zhao23] from the (i) dynamical systems literature, the “(system) identification” refers to the estimation of the underlying dynamical system, without any theoretical analysis or guarantees on whether the estimated quantities recover the ground-truth latents, transition and generative functions. In contrast, in the (ii) nonlinear ICA literature, the term “identification” encapsulates not only the estimation but also a theoretical framework showing when the estimated model recovers the ground-truth generative model (up to some indeterminacy transformation classes) [HyvarinenPajunen99; Hyvarinen+23].
>
> In our work, we use the term “identifiability” with the latter meaning from the (ii) nonlinear ICA literature, to refer to our theoretical framework which guarantees that the ground-truth factors and functions are recovered under certain assumptions. In this context, to the best of our knowledge, our work is the "the first framework that allows for the identification of the unknown transition function alongside latent states and the generative function". While the distinction becomes clear from the theoretical definitions in Sec. 2.2-2.3, we added a footnote to the contributions paragraph (ln. 106-107) to clarify already in the Introduction any possible confusion.
>
> To clarify the distinction further, let us use ICA and PCA, i.e., common latent variable models, as examples, and see what they tell us about the latent variable identification in the mentioned works, e.g., [Sussillo16, Schimel21]. Under the assumption of non-Gaussianity, ICA recovers ground-truth factors up to an elementwise scaling. On the other hand, PCA fails to do so and recovers the factors up to a rotation indeterminacy (linear mixing) due to its standard normal prior. As the mentioned works [Sussillo16, Schimel21] also use a standard normal prior for the latent variables, they suffer from the same rotation indeterminacy even in the case of linear emission functions. When we allow for nonlinear emission functions, the non-identifiability gets more severe, i.e., there exist infinitely many function classes leading to latent estimations that are equivalently valid for the probabilistic model, as shown theoretically in [HyvarinenPajunen99] and empirically in [Locatello+19].
>
> [Sussillo16]: LFADS - Latent Factor Analysis via Dynamical Systems
>
> [Schimel21]: iLQR-VAE : control-based learning of input-driven dynamics with applications to neural data
>
> [Hess23]: Generalized Teacher Forcing for Learning Chaotic Dynamics
>
> [Zhao23]: Physics-informed deep sparse regression network for nonlinear dynamical system identification
>
> [Locatello+19]: Challenging Common Assumptions in the Unsupervised Learning of Disentangled Representations
>
> ***W2) The generality of the proposed framework.***
>
> We would like to clarify the following points:
> * We emphasize that our theoretical framework is general, i.e., not tailored for a specific application, and our generative model is not limited to robotics or RL, but encompasses any latent dynamical system with high-dimensional observations.
> * We chose three datasets with different characteristics to keep our experimental setup (i) as general as possible, and (ii) comparable with the related works that are closest to ours (LEAP [Yao+21] and TDRL [Yao+22]). In the following, we briefly describe our three datasets, noting that only the second one comes from a dynamical system often used in RL: (i) a synthetic dataset satisfying our modeling assumptions as common in nonlinear ICA literature [Hyvarinen+16,Hyvarinen+17,Hyvarinen+19,Yao+21,Yao+22], (ii) a cartpole dataset with high-dimensional video observations whose low-dimensional ground-truth dynamics are governed by a set of nonlinear differential equations [Yao+22], and (iii) the high-dimensional (50-dim) real-world Mocap dataset with longer training and prediction horizons (ranging between 75-300) [Yao+21,Yao+22,Yildiz+19,Li+20,Auzina+23].
>
> This said, we changed the first sentence of the Abstract as follows: “This work aims to recover the states and their time evolution in a latent dynamical system from high-dimensional sensory measurements” (ln. 11-12), and made our statement more specific in a later sentence as follows: “We empirically demonstrate that it improves generalization and interpretability of target dynamical systems” (ln. 22-23).

---

> > ### Author Response · Authors · 2024-11-21
> >
> > ***Q1) Latent dimensionality***
> >
> > Our theory assumes that the dimensionality $K$ of the latent space and noise is known.
> > In practice, selecting the number of components in linear and nonlinear ICA is an open question with possible proposed methods such as bootstrapping [Himberg+04], qualitative analysis (visualization) [Himberg+04], information-theoretic criteria [Krumsiek+12], cross-validation [Bro+08] and downstream task performance [Zhu+23].
> >
> > In terms of latent dimensionality selection, one advantage of our model compared to a standard nonlinear ICA method is that our model can make future predictions, which can be used to cross-validate different hyperparameter configurations. To make reasonable predictions, the model should recover the underlying dynamical structure, which would not be possible with too few or too many components.
> >
> > In our experiments, we chose the dimensionality using domain knowledge and cross-validation. Specifically, we chose
> > - $K=8$ for the synthetic experiment by domain knowledge, which is equal to the number of ground-truth factors for the synthetic experiment, similarly to the related works [Hyvarinen+16,Hyvarinen+17,Hyvarinen+19,Khemakhem+20,Yao+21,Yao+22],
> > - $K=8$ for the cartpole experiment, by cross-validation,
> > - $K=3$ for the Mocap dataset, by cross-validation.
> >
> > [Himberg+04]: Validating the independent components of neuroimaging time series via clustering and visualization
> >
> > [Bro+08]: Cross-validation of component models: a critical look at current methods
> >
> > [Krumsiek+12]: Bayesian independent component analysis recovers pathway signatures from blood metabolomics data
> >
> > [Zhu+23]: Unsupervised representation learning of spontaneous MEG data with nonlinear ICA
> >
> > ***Q2) Sparse noise***
> >
> > We assume ‘sparse’ noise here refers to either that (i) the number of nonzero values in the process noise $\mathbf{s}$ is bounded or (ii) the transition function $\mathbf{f}(\mathbf{z}_{t-1}, \mathbf{s}_t)$ only depends on a small subset of its inputs. It would be straightforward to include sparsity priors in our framework on latent factors, the process noise or the transition function inputs, although we do not consider it in the paper. As an example, a related work (LEAP [Yao+21]) models the transition function depending only on a sparse set of inputs by a masking layer.
> >
> > There is also a recent line of work on the intersection of (mechanism) sparsity and identifiability [Zheng+22, Lachapelle+23, Xu+24]. We think that it is an interesting future direction to investigate the comparisons and synergies between our assumptions and sparsity assumptions.
> > We hope this answers your question, please let us know if there is need for further clarification.
> >
> > [Zheng+22]: On the Identifiability of Nonlinear ICA: Sparsity and Beyond
> >
> > [Lachapelle+23]: Synergies between Disentanglement and Sparsity: Generalization and Identifiability in Multi-Task Learning
> >
> > [Xu+24]: A Sparsity Principle for Partially Observable Causal Representation Learning

---

> > > ### Comment · Reviewer_oxFU · 2024-11-23
> > >
> > > Thank the authors for addressing my concerns and questions. I've decided to raise the score.

---

> > > > ### Author Response · Authors · 2024-11-26
> > > >
> > > > We are pleased to hear that our reply addressed your concerns well. Thank you once again for your interesting comments!

---

### Official Review · Reviewer_iU39 · 2024-11-01

**Soundness:** 3
**Presentation:** 2
**Contribution:** 3
**Rating:** 6
**Confidence:** 3

**Summary:**

This work deals with the general problem of identifying (i.e. inferring) the nonlinear transition function underlying the dynamics of complex dynamical systems. The authors extend recent theoretical work on identifiability of latent states in dynamical systems to the identifiability of their transition function. Furthermore, the authors provide a neural-based inference model that implements the key aspects of the theory and relies on neural variational inference. The authors then test their methodology on three settings: (i) a synthetic dataset that mirrors their generative model; (ii) a cartpole simulation which is represented as a video signal; and (iii) three sets of experimental motion capture data. The results demonstrate the proposed models outperform the baselines, not only on the identification of the latent variables, but also on forecasting tasks.

**Strengths:**

- This work tackles a key problem of (nonlinear) dynamical system modelling, namely the identifiability of the transition function underlying the dynamics;
- The authors naturally extend the identifiability theory from previous works to the identification of the nonlinear transition function of the hidden process. Their proofs are well written and easy to follow;
- The authors demonstrate that their model implementation of their theory outperforms different baselines, in different settings. The results are very convincing (see however the questions below).

**Weaknesses:**

In my view, the main weaknesses of the paper have to do with its presentation. The three most pressing points are:

1. There is somewhat of a disconnect between the theory section and the practical implementations, which I think could easily be improved. Especially when it comes to the experimental section. Questions 2 – 8 below point to concepts or ideas which are not clearly explained, that could be explained better or that are simply absent. In particular, it is not clear to me how the auxiliary ($\mathbf{u}$) and noise ($\mathbf{s}$) variables are used (or represented) in the experiments. To improve the readability of the paper, I'd suggest the authors explicitly describe the role of $\mathbf{u}$ and $\mathbf{s}$ in each experiment.

2. The target datasets could be explained and motivated better. It is not clear from a first reading why these datasets are relevant in the context of the model (see e.g. questions 3 and 4 below). This point is clearly related to point 1 above. I believe that if the authors explain what $\mathbf{u}$ and $\mathbf{s}$ represent in each dataset, it will become clearer why those target datasets are interesting.

3. The paper builds on a large collection of previous works which deal with identifiability and go all the way back to independent component analysis (ICA). The authors assumed the reader is versed in the development of ICA, from its linear to its nonlinear versions, and simply present e.g. the augmented dynamics (line 126) and the auxiliary variable (line 130), or the identifiability assumption, without any further explanations and context. There is however some nice physical interpretation underlying e.g. the conditional independence assumption of the noise process given the auxiliary variable, which can be traced back to Packard et al. [1980], and that is also nicely explained (albeit in a different sense) by e.g. Hyvarinen et al. [2019]. Even in the introduction, lines 41-49, the authors could already mention that their work builds on classical ideas of ICA. In short, I think the authors could better exploit the history of nonlinear ICA and its application to dynamical systems to better motivate their proposal.

Note:  *I will happily increase my score once the authors answer the questions below* and explain how they will try to incorporate this information back into the paper.

**Other comments and/or typos**
- in line 88 the authors write *the generative function*. Do the authors mean the function $\mathbf{g}$? I’d suggest, for the sake of clarity, to either stick to the “instantaneous mapping” label used later in line 112, or better use the more standard “emission model” label.
- in your KL loss term (lines 290 inside Algorithm 1) you write that the posterior over the noise $\mathbf{s}$ is conditioned on $\mathbf{u}$. This is a typo, isn’t it?

**References**
- Packard et al. [1980]: Geometry from a time series
- Hyvarinen et al. [2019]: Nonlinear ICA Using Auxiliary Variables and Generalized Contrastive Learning

**Questions:**

1. Is there a reason why the noise posterior $q(\mathbf{s}\_{t} | \mathbf{\tilde x}, \mathbf{z}\_{t-1} )$ explicitly depends on $\mathbf{z}\_{t-1}$? Does this follow from your theory? If so, how?

2. In line 140 the authors write “setting $\mathbf{u}$ to an *observed regime index*...” What do you mean by observed regime index? Do you mean the time index $t$ of the observation $\mathbf{x}(t)$? Or do you mean the observation value $\mathbf{x}(t)$ itself?

3. How is $\mathbf{u}$ used in your practical implementation? Put another way, do you set $\mathbf{u}$ to be an *observed regime index* or $\mathbf{s}\_{t-1}$? Or is it only implicitly define via the prior $p(\mathbf{s}\_t| \mathbf{u})$? In Figure 6, $\mathbf{u}$ is shown as an input to the prior noise model, which is used for forecasting. Can you please explain how you choose $\mathbf{u}$ in each experiment?

4. What does $\mathbf{s}$ represent in the synthetic experiments? Similarly, what does it represent in the Cartpole dataset?

5. In Table 1, why is $\text{MCC}[\mathbf{z}] > \text{MCC}[\mathbf{s}]$ for all models? Is the noise variable harder to infer in practice?

6. How do you interpret the inferred $\mathbf{s}$ in the MOCAP datasets?

7. In the synthetic dataset description of the appendix (lines 1235, 1236) the authors write: the number of environments is R = 20. What do these environments represent?

8. Are the sufficient variability assumptions (A3, A4) satisfied in the synthetic and Cartpole experiments? If so, can you please explain why?

9. Why are the simulated trajectories (of the synthetic dataset) so short? Meaning: what is the motivation behind such a setting?

---

> ### Author Response · Authors · 2024-11-21
>
> ***W1: Role of u and s and in each experiment. -> Questions 2-8.***
>
> Thank you for raising this point! In the practical implementations and experiments, we assume the data consists of piecewise stationary time series, and each stationary period has a "regime label", i.e., an indicator to select the generative model for the period from a set of alternative distributions. The auxiliary variable $\mathbf{u}$ denotes this regime label. The noise $\mathbf{s}$ represents random or unmodeled external variations in the process, not captured by deterministic dynamics. Below, we answer questions 2-8 explicitly. We combine this information and incorporate them to the main text in the **Datasets** paragraph of Section 4 between ln. 368-387.
>
> ***W2: Why did we pick these datasets? -> Questions 3–4.***
>
> We chose three datasets with different characteristics to keep our experimental setup (i) as general as possible, and (ii) comparable with the related works that are closest to ours (LEAP [Yao+21] and TDRL [Yao+22]). Our datasets include: (i) a synthetic dataset satisfying our modeling assumptions as common in nonlinear ICA literature [Hyvarinen+16,Hyvarinen+17,Hyvarinen+19,Yao+21,Yao+22], (ii) a cartpole dataset with high-dimensional video observations whose low-dimensional ground-truth dynamics are governed by a set of nonlinear differential equations [Yao+22], and (iii) the high-dimensional (50-dim) real-world Mocap dataset with longer training and prediction horizons (ranging between 75-300) [Yao+21,Yao+22,Yildiz+19,Li+20,Auzina+23].
>
> Our motivation for the synthetic experiment is to validate our theory, and for the cartpole experiment is to show our model’s ability to recover low-dimensional latent dynamics from a different kind of high-dimensional input modality (videos). To test our model’s capability to predict longer sequences of complex real-world dynamics, we also included the MOCAP experiment.
>
> ***W3: Connections between our work and (i) classical ideas of ICA, (ii) history of nonlinear ICA and its applications to dynamical systems, and (iii) physical interpretation underlying the conditional independence assumption in [Packard+80] and [Hyvarinen+19]***
>
> Thanks for the insightful comment! We agree that our presentation can improve by highlighting these connections. In the following, we discuss them and how we incorporate these into the text:
> * **Independence assumption (classical idea from linear ICA):** In lines 45-46, we added: “our approach builds on the classical ideas of ICA, namely, the assumption of independent components.”
> * **Application of ICA to dynamical systems:** In lines 46-47, we added: “In the case of a linear emission function, this independence assumption has been used to identify states in dynamical systems [Ciaramella+06, Kerschen+07]”. We are not aware of any works applying nonlinear ICA to dynamical system learning except LEAP [Yao+21] and TDRL [Yao+22], which we mention in the paper (if you have further pointers, we would be happy to discuss them in the paper).
> * **Conditional independence in nonlinear ICA:** We elaborate on what the conditional independence assumption implies and its connection to [Packard+80] in lines 85-90: “To resolve non-identifiability, inspired by nonlinear ICA, we introduce additional stochastic variables that are conditionally independent of each other given auxiliary variables. This conditional independence assumption implicitly encodes new, external information that is sufficiently independent of the observations [App. E, Hyvarinen+19]. Interestingly, the idea of conditioning on auxiliary variables has been proposed in other contexts, e.g., [Packard+80] showed that time-delayed auxiliary variables help resolve underlying data generating dynamics of chaotic systems from the residual independent noise.”
>
> [Ciaramella+06]: ICA based identification of dynamical systems generating synthetic and real world time series
>
> [Kerschen+07]: Physical interpretation of independent component analysis in structural dynamics
>
> ***C1: emission***
>
> Thanks for pointing this out. We changed all the relevant terms to “emission function”.
>
> ***C2: Clarify KL term***
>
> Yes, it is a typo. We actually simplify the conditioning sets by omitting $\mathbf{u}$, as adding them didn’t give any performance improvements. We fixed the conditioning sets in Algorithm 1.
>
> ***Q1: Clarify noise posterior.***
>
> In the graphical model $\mathbf{x}_t$ is a descendant of both $\mathbf{s}_t$ and $\mathbf{z}(t-1)$ (through $\mathbf{z}_t$). Hence, $\mathbf{s}_t$ and $\mathbf{z}(t-1)$ are dependent conditionally on $\mathbf{x}$ and therefore we condition the posterior of $\mathbf{s}_t$ with $\mathbf{z}(t-1)$.

---

> > ### Author Response · Authors · 2024-11-21
> >
> > ***Q2: Setting $u=r$ observed regime index***
> >
> > We mean setting the auxiliary variable $\mathbf{u}$ to a regime (or environment) label, $r=1, …, R$, where the data is observed under $R$ distinct regimes (or environments). Here, the process noise is assumed to show nonstationarity behavior between distinct regimes, as in TCL [Hyvarinen+16]. For a visual example, see Fig. 1 in TCL [Hyvarinen+16].
> > As an example in the context of the cartpole system, we can consider $R$ distinct observation regimes with different gravity, mass or wind conditions, among which the distribution of latents and process noise is sufficiently different.
> > To clarify the confusion around the word ‘index’, we changed the term “regime index” to “regime label” and the term “time index t” to “time point t” throughout the text.
> >
> > ***Q3: The auxiliary variable $\mathbf{u}$ in the implementation and the experiments***
> >
> > In the implementation and the experiments, we set the auxiliary variable $\mathbf{u}$ to the regime label where the data is observed under distinct regimes. We incorporated the previous sentence to the beginning of Section 3, ln. 259-261. We leave the practical scenarios of setting it to $\mathbf{s}_{t-1}$ for future work. In the implementation, the auxiliary variable $\mathbf{u}$ is used in the prior $p(\mathbf{s}_t | \mathbf{u})$ which is modeled as a conditional normalizing flow.
> >
> > In the experiments, we observe $(\mathbf{x}_t, \mathbf{u}_t)$ pairs where each observation $\mathbf{x}_t$ has an associated regime label $\mathbf{u}_t$. For the synthetic experiment, we have $R=20$ regimes. For the cartpole experiment, we have $R=6$ regimes (5 for training, 1 for testing). For the MOCAP experiment, the different regimes correspond to different persons walking. In MOCAP-SINGLE, we have just one person, hence $R=1$, and $\mathbf{u}_t$ is constant, demonstrating a case violating our assumptions. In MOCAP-MULTI/SHIFT, we have 6 persons, hence $R=6$. For each observation $\mathbf{x}_t$, the model (or the prior $p(\mathbf{s}_t | \mathbf{u})$) takes $\mathbf{u}_t$, i.e., the one-hot encoded regime label, as input.
> >
> > ***Q4: What does $\mathbf{s}$ represent in synthetic and cartpole experiments?***
> >
> > The noise $\mathbf{s}$ represents random or unmodeled external variations in the process, not captured by deterministic dynamics. In the synthetic experiment, we did not assign physical interpretations to the variables, and the distribution of $\mathbf{s}$  simply changes between distinct regimes to provide non-stationarity to satisfy the assumptions.
> >
> > In the cartpole experiment, the ground-truth latent dynamical system is deterministic, governed by a set of nonlinear differential equations. So, here the noise $\mathbf{s}$ accounts for the variability due to unknown initial conditions and different gravity values between the environments. The gravity values are {5, 10, 20, 30, 40, 90} for regimes {A, B, C, D, E, F}. The regime labels are non-informative and observed, while the gravity values are unobserved.
> >
> > ***Q5: Is noise harder to estimate?***
> >
> > In our generative model, observations $\mathbf{x}(t)$ directly depend on $\mathbf{z}(t)$: $\mathbf{x}(t) = \mathbf{g}(\mathbf{z}(t))$, while the dependency to $\mathbf{s}(t)$ is indirectly through $\mathbf{z}(t)$: $\mathbf{x}(t) = \mathbf{g}( \mathbf{f}(\mathbf{z}_{t-1}, \mathbf{s}_t) )$. To reconstruct observations $\mathbf{x}(t)$, the model has to recover $\mathbf{z}(t)$ without having to fully identify the noise $\mathbf{s}(t)$. So, the noise is harder to estimate, to the best of our understanding.
> >
> > ***Q6: Interpretation of $\mathbf{s}$ in MOCAP***
> >
> > In the MOCAP experiment, we considered walking trials of distinct subjects as distinct regimes $u=r$. With this choice, we interpret the process noise $\mathbf{s} | \mathbf{u}$ modeling the external variations in the observed data, which might show nonstationary behavior due to subject-specific characteristics such as limb lengths, joint angles, sensor positionings, sensor noise, or initial conditions.
> >
> > ***Q7: Synthetic environments.***
> >
> > In the synthetic experiment, environments are different regimes as identified by the regime label, and practically they specify the observation periods with varying noise distributions. See also our answer to Q4.
> >
> > ***Q9: Short synthetic trajectories.***
> >
> > We chose the synthetic training sequences as $T_o+T_{input} = 6$ as in LEAP [Yao+21] and TDRL [Yao+22], to keep our setup comparable to the two closest related works. Empirically, they form a useful synthetic environment to validate our theory. To present our model’s performance on longer sequences, we have included results on the MOCAP data sets which have training sequences of lengths 75 and 300, and test prediction horizons of 75 and 297.

---

> > > ### Author Response · Authors · 2024-11-21
> > >
> > > ***Q8: Sufficient variability assumptions (A3, A4) in synthetic and cartpole experiments.***
> > >
> > > In the synthetic experiment, we use noise with (randomly) modulated variance between environments. Under this modulated noise, sufficient variability assumptions (A3, A4) hold.
> > >
> > > We will briefly add the analysis for the noise $\mathbf{s}$ (assumption A4) here. The analysis for $\mathbf{z}$ (assumption A3) is similar. We have incorporated this discussion to Appendix F.2 and we have clarified the ground-truth data generating process in Appendix F.1 - **Synthetic Dataset** paragraph.
> > >
> > > Our ground-truth functions for this experiment are:
> > > $$
> > > 	s_{kt} = \frac{1}{a_k(\mathbf{u})} \epsilon_t, \qquad \epsilon \sim N(0,1),
> > > $$
> > > $$
> > > 	z_{kt} = f_k(\mathbf{z}(t-1), \mathbf{z}(t-2)) + s_{kt},
> > > $$
> > >
> > > The log probability is: $\log p(s_{kt} | \mathbf{u}) = q_{kt} = - \log 2 \pi + \log a_k - \frac{a_k^2}{2} s_{kt}^2$. Then, the required partial derivatives take the following form:
> > > $$
> > > \frac{\partial^2 q_{kt}}{\partial s_{kt} \partial u_l } = -2 a_k(\mathbf{u}) \frac{\partial a_k}{\partial u_l}(\mathbf{u}) s_{kt}
> > > $$
> > > $$
> > > \frac{\partial^3 q_{kt}}{\partial s_{kt}^2 \partial u_l } = -2 a_k(\mathbf{u}) \frac{\partial a_k}{\partial u_l}(\mathbf{u})
> > > $$
> > > These two equations tell us that if the $2K$ functions $- a_k(\mathbf{u}) \frac{\partial a_k}{\partial u_l}(\mathbf{u}) s_{kt}$ and $- a_k(\mathbf{u}) \frac{\partial a_k}{\partial u_l}(\mathbf{u})$ for $k = 1, …, K$ in $\mathbf{u}$ are linearly independent, then Assumption (A4) holds.
> > >
> > > For our synthetic setup, we choose the modulator variables $a_k(\mathbf{u})$ uniformly random between $[0,1]$. To simplify the analysis, let us assume that we have $a_k(\mathbf{u}) =\sqrt{2 u_l} w_k$ with $w_k \sim U(0,1)$. Then, the $2K$ functions become $- w_{k} s_{kt}$ and $ - w_k$, which are random functions. This results in a $2K \times 2K$ matrix with random entries in the form of $ - w_k$ and $- 2 w_{k} s_{kt}$ with $w_k \sim U(0,1)$ and $s_{kt} \sim \mathcal{N}(0, a_k^{-2})$. We conjecture that the probability of these random vectors to be linearly independent is 1. To validate this empirically, we sampled 10000 $2K \times 2K$ matrices with entries $- 2 w_{k} s_{kt}$ and $ - w_k$, where $w_k \sim U(0,1)$ and $s_{kt} \sim N(0, w_k^2)$. They all had rank $2K$.
> > >
> > > While we simulate cartpole data under different gravity values to create nonstationarity, Assumptions (A3, A4) do not exactly hold in the cartpole experiment since the nonlinear differential equations underlying the cartpole system are deterministic.

---

> ### Comment · Reviewer_iU39 · 2024-11-24
>
> Thanks very much for the very complete response. I have increased my score.
>
> Could the authors also include the responses to Q2-Q9 somehow into the manuscript? Maybe as an Appendix, that you reference here and there inside the main text.

---

> > ### Author Response · Authors · 2024-11-26
> >
> > Thank you for your response! We are pleased to hear that our reply addressed your concerns completely. We have included the responses to the Q2-Q9 and updated the manuscript:
> > * Q2: To the main text (lines 150–153).
> > * Q3 (implementation of u): To the main text (lines 285–286).
> > * Rest (Q3-Q9): To the Appendix E and F, which we referenced from the main text.
> >
> > Thank you once again for your insightful comments and questions, which have helped us improve our manuscript!

---

### Official Review · Reviewer_XAHn · 2024-11-02

**Soundness:** 3
**Presentation:** 3
**Contribution:** 3
**Rating:** 8
**Confidence:** 4

**Summary:**

In this paper, the authors introduce a theory under which one can identify both the latent states as well as the latent dynamics from observational data (up to permutations and invertible transformations). Next, the authors introduce a practical algorithm, based on sequential variational autoencoders, for learning identifiable state-space models. Experiments provide empirical evidence that proposed approach can correctly recover the latent states and the latent dynamics.

**Strengths:**

My favorite thing about this paper is the simplicity of the proposed approach. By just simply changing the form of the latent dynamics model, i.e., as opposed to additive noise, which is the standard, the noise is an input to the transition function, and assuming a dimension-dependent dynamics function, one can recover the states and dynamics (under some assumptions). I think this is a massive break through for people who learn state-space models.

**Weaknesses:**

While the paper is strong, there a couple of minor weaknesses. Firstly, is that assumption 2 doesn't seem to align with one of the motivating examples in the introduction, which is model-based RL. Specifically, assumption 2 does not allow for process noise/control values that are temporally correlated. But in model-based RL, the process noise *is* temporally correlated because the control value chosen at time $t$ depends on the previous state of the system. I found this contrast jarring.

Second, the experiments section is great but I think it would be great to have a seqVAE baseline that is more similar to proposed generative model. In the experiments section, the baseline seqVAE is the KalmanVAE, which is a hierarchical SSM. While the proposed approach outperforms KalmanVAE, its not clear whether it outperforms it is truly better or if there is a model mismatch between the true generative model and the one used by KalmanVAE. I think training a seqVAE using the generative model as described in section 2.1 would strengthen the results.

Also, section 4.4 doesn't make sense at a first glance. If I had the true latents then training a model in a supervised fashion should be better than learning both in an unsupervised fashion. When doing the comparison, were the number of parameters matched? I ask because if your latent state has $K$ dimensions then one would need $K$ neural networks, so it might be the case that the proposed approach is working just because the model has more expressive power.

Lastly, I think the proofs in the appendix could benefit with a little more hand-holding to follow along!

**Questions:**

1. Why was such a small prediction horizon used for testing? I found this odd for the synthetic data.
2. I see that training sequences used to train the model are relatively short. Is there a reason why?

---

> ### Author Response · Authors · 2024-11-21
>
> ***W1) Assumption 2 vs. motivating example:***
>
> Thanks for the insightful comment! In our framework, the process noise $\mathbf{s}$ or the auxiliary variable $\mathbf{u}$ differs from control variables. The process noise $\mathbf{s}$ represents random or unmodeled external variations in the process, which might show autocorrelation or nonstationarity, and these are not captured by deterministic dynamics. The auxiliary variable $\mathbf{u}$ refers to different regimes, e.g., cartpole environments with different pole lengths or gravity values. Both the process noise $\mathbf{s}$ and the auxiliary variable $\mathbf{u}$ are set independently of the process history. For different regimes $\mathbf{u}$, you can still incorporate controls in addition to the process noise, in which case $\mathbf{u}$ would also reflect the differences in the policies between the regimes. The cartpole experiments show that our framework almost fully recovers underlying latents when the process noise and control signals influence the transitions together.
>
> We realized that we interpreted the process noise in the main text as (temporally uncorrelated) control signals in lines 118-120. We removed this interpretation to clarify this point, and added the following interpretation of the process noise (in lines 128-129): “random or unmodeled external variations in the process, not captured by deterministic dynamics”.
>
> ***W2) SeqVae baseline:***
>
> We present an ablation study in the synthetic dataset in Appendix E.1, where we compare our full model with a sequential VAE directly using the generative model described in section 2.1, in terms of MCC[z] and future MSE.
>
> ***W3) Section 4.4: 2-stage vs. 1-stage:***
>
> Thank you for the nice catch! We ran the experiment also for $K$-MLP transition functions for the 2-stage model with the same data. We updated Figure 5. The results of transition functions with $K$-MLPs and a single MLP for the 2-stage model are very similar. This is because 1-MLP transition function is already expressive enough: the number of layers in the learned model (#layers=3) is larger than the number of layers in the data-generating random MLP (#layer=2).
>
> ***W4) More readable proofs in the appendix:***
>
> Thanks for pointing this out. We have added clarifications to better connect the equations in the proofs in the Appendix C.1 and C.2.
>
> ***Q1) Small prediction horizon used for testing for the synthetic data.***
>
> For the synthetic data, the training sequences have $T_o+T_{dyn} = 2+4 = 6$ steps, i.e., they are as long as in LEAP [Yao+21] and TDRL [Yao+22], to keep our setup comparable to the two closest related works. We chose the prediction horizon $T_{future} = 8$ for testing as the double of the dynamical steps $T_{dyn}=4$ seen during the training. To test our model performance on longer prediction horizons, we have included the MOCAP experiments, where the prediction horizons range from 75 to 300 time steps.
>
> ***Q2) Short training sequences.***
>
> There is no particular computational or statistical restriction for the sequence length (see computational analysis in the answer for Reviewer i2e2), and we set them similarly to previous work [Yao+21,Yao+22]. They could also be considerably longer as demonstrated by the MOCAP experiment.

---

> > ### Comment · Reviewer_XAHn · 2024-11-22
> > **Response**
> >
> > Thanks for the response. After reading your response, I realized that the current framework would still work in the RL regime. If I was doing RL, then I would not have a fixed $u$ but a sequence of $u_{1:T}$, which would be temporally correlated. Crucially, the current frameworks assumes that the process noise is conditionally independent given the $u$, which would still work even if there was a sequence.

---

> > > ### Author Response · Authors · 2024-11-26
> > >
> > > We are glad to hear that your concern on the temporally correlated $\mathbf{u}_{1:T}$ has been addressed. Thank you once again for your insightful and thoughtful feedback, which has contributed to improving our manuscript!

---

### Official Review · Reviewer_i2e2 · 2024-11-02

**Soundness:** 3
**Presentation:** 3
**Contribution:** 3
**Rating:** 6
**Confidence:** 2

**Summary:**

This paper proposes a structure with a theoretical foundation for identifying latent dynamics under certain assumptions. The implementation is standard. The experiments' results show various advantages in different scenarios, even when the assumptions are violated.

**Strengths:**

In Table 3, the author(s) claim, "Our approach is the first to predict future states of a dynamical system by utilizing a transition function that identifies the underlying, true dynamics." As far as I know, many approaches can do similar things without a theoretical foundation for the identification.

**Weaknesses:**

Perhaps a theoretical or practical time complexity analysis would be more appropriate.

**Questions:**

1. Does the identifiability still exist when I have any external input, like the control input in the state-space model?
2. What's the exact definition of $s_t$, and what's the motivation for incorporating $s_t$ into the model?
3. (minor) For Table 1, do you use any stats test to confirm your approach is the best?

---

> ### Author Response · Authors · 2024-11-21
>
> ***W1) Time complexity analysis:***
>
> Thanks for raising this interesting point. The computational complexity of our algorithm is equivalent to any sequence model, e.g., RNN. For each forward pass, we encode the latents sequentially in time. The complexity is linear in the input sequence length $\mathcal{O}(T)$. We agree that a time complexity analysis would enrich the manuscript. So, we added this to lines 276-277.
>
> ***Q1) Identifiability w/ external input:***
>
> Thanks for the comment. The identifiability proof requires that the augmented dynamics function $\mathbf{f_{\texttt{aug}}}: \mathbb{R}^{2K} \rightarrow \mathbb{R}^{2K}$ is bijective. The identifiability proof continues to hold as long as the augmented dynamics function conditioned for a given control input $a$, $\mathbf{f}_{\texttt{aug}}(\cdot | a): \mathbb{R}^{2K} \rightarrow \mathbb{R}^{2K}$, is bijective. We have added this interesting point as Remark 2 in Appendix C (ln. 798-802).
>
> ***Q2) Motivation and definition for $\mathbf{s}_t$:***
>
> The noise $\mathbf{s}$ represents random or unmodeled external variations in the process, not captured by deterministic dynamics. This external randomness can result from environmental conditions such as wind or weather conditions acting on the system, surface imperfections, sensor noise, unknown initial conditions, etc.
>
> Such inherent stochasticity have been reported for real-world dynamical systems in different domains such as video dynamics [DentonFergus18; Franceschi+20], and brain activity (spike trains and MEG) data in neuroscience [Sussillo+16, Schimel+21, Hyvarinen+16]. As a concrete example in the context of neural spiking trains, they can be physically interpreted as stochastic inputs to the recorded neurons received from the unrecorded neurons [Sussillo+16]. These works have shown empirically that the prediction results improve when their model has taken this stochasticity into account.
>
> [DentonFergus18]: Stochastic Video Generation with a Learned Prior
>
> [Franceschi+20]: Stochastic Latent Residual Video Prediction
>
> [Sussillo+16]: LFADS - Latent Factor Analysis via Dynamical Systems
>
> [Schimel+21]: iLQR-VAE : control-based learning of input-driven dynamics with applications to neural data
>
> [Hyvarinen+16]: Unsupervised Feature Extraction by Time-Contrastive Learning and Nonlinear ICA
>
> ***Q3) Statistical tests for Table 1:***
>
> Based on Welch’s t-test, our method is consistently best (p<0.01) for all but one comparison; only comparing the MCC[z] score in the synthetic experiment against LEAP-NP gives p=0.09 – we should have bolded the LEAP-NP result as well and now changed the manuscript accordingly.

---

> > ### Comment · Reviewer_i2e2 · 2024-11-25
> >
> > Thanks for addressing all my questions. I've decided to raise the score to 6.5.

---

> > > ### Author Response · Authors · 2024-11-26
> > >
> > > We are pleased to hear that our reply addressed your concerns well. Thank you once again for your interesting comments and questions, which have helped us improve our manuscript!

---

### Meta-Review · Area_Chair_jSdc · 2024-12-18

**Metareview:**

This paper provides a theoretical state-space modeling framework for identifying both latent states and nonlinear latent dynamics, in contrast to previous works that assumes linear transitions. It also introduces a practical algorithm based on variational autoencoder and shows superior performance in recovering the latent dynamics, making accurate future prediction and adapting to new environments.

All reviewers appreciate its theoretical foundation to the proposed framework and a practical algorithm. Experiments also show robust performance across diverse datasets and strong comparison to baselines.

**Additional Comments On Reviewer Discussion:**

The reviewers have various concerns in their initial reviews, including:
- The conditional independency assumption for control signals
- Lack of comparison to a generative model baseline
- Lack of time complexity analysis
- Disconnection between the theory section and the practical implementations
But the authors' rebuttal addressed all the concerns satisfyingly. The reviewers either maintained their positive rating or raised the rating after the rebuttal.

---

### Decision · Program_Chairs · 2025-01-22

Accept (Poster)